# Transient expression of an adenine base editor corrects the Hutchinson-Gilford progeria syndrome mutation and improves the skin phenotype in mice

Daniel Whisenant[1,3], Kayeong Lim [2,3], Gwladys Revêchon [1], Haidong Yao[1], Martin O. Bergo [1], Piotr Machtel[1], Jin-Soo Kim [2] & Maria Eriksson [1✉]

Hutchinson-Gilford progeria syndrome (HGPS) is a rare premature ageing disorder caused by a point mutation in the *LMNA* gene (*LMNA* c.1824 C > T), resulting in the production of a detrimental protein called progerin. Adenine base editors recently emerged with a promising potential for HGPS gene therapy. However adeno-associated viral vector systems currently used in gene editing raise concerns, and the long-term effects of heterogeneous mutation correction in highly proliferative tissues like the skin are unknown. Here we use a non-integrative transient lentiviral vector system, expressing an adenine base editor to correct the HGPS mutation in the skin of HGPS mice. Transient adenine base editor expression corrected the mutation in 20.8-24.1% of the skin cells. Four weeks post delivery, the HGPS skin phenotype was improved and clusters of progerin-negative keratinocytes were detected, indicating that the mutation was corrected in both progenitor and differentiated skin cells. These results demonstrate that transient non-integrative viral vector mediated adenine base editor expression is a plausible approach for future gene-editing therapies.

[1] Department of Biosciences and Nutrition, Center for Innovative Medicine, Karolinska Institutet, Huddinge, Sweden. [2] Center for Genome Engineering, Institute for Basic Science (IBS), Dajeon, South Korea. [3]These authors contributed equally: Daniel Whisenant, Kayeong Lim. ✉email: maria.eriksson.2@ki.se

Hutchinson-Gilford progeria syndrome (HGPS) is a premature aging disorder affecting 1 in 18 million children worldwide[1]. Within the first years of life, affected children develop several pathologies, which are commonly observed in ageing, including alopecia, osteoporosis, atherosclerosis and vascular calcifications. Cardiovascular complications and stroke are the most common causes of death and the median survival is 14.6 years[2,3]. The first observed symptoms in HGPS patients are failure to thrive and scleroderma-like skin changes. Affected children display clinical characteristics such as epidermal thickening, hyalinization and disorganization of collagen bundles, reduced numbers of hair follicles and sebaceous glands with a thin or absent hypodermal layer[2,4–13].

The majority of HGPS cases are caused by an autosomal dominant point mutation within the *LMNA* gene (c.1824C > T, p. G608G), which amplifies a cryptic splice event in exon 11. The resulting mis-splicing leads to an internal deletion of 50 amino acids of the prelamin A precursor protein[14,15]. This deletion removes the recognition site for the metalloproteinase *ZMPSTE24* in the prelamin A precursor protein and leads to the production of a truncated protein called progerin, which remains permanently farnesylated and carboxymethylated at the carboxyl terminus[8,16]. Progerin accumulation within the nuclear lamina interferes with the nuclear structure, architecture and macromolecular interactions, which altogether impair nuclear function. Progerin accumulation also leads to nuclei lobulation, loss of heterochromatin, lamina thickening as well as clustering of nuclear pores which interferes with mitosis[14,17–19].

Recently, inhibiting farnesylation with a farnesyltransferase inhibitor became an FDA-approved treatment strategy for HGPS[3,20,21]. Current CRISPR-Cas9 therapeutic approaches have been developed to interfere with *LMNA* and progerin expression[22,23]. With the development of CRISPR-Cas9 based single nucleotide editing technologies such as adenine base editors (ABEs) it has been shown that A:T base pairs can be converted to G:C base pairs at a high frequency, without introducing DNA double-strand breaks, demonstrating a low off-target editing profile[24]. With the use of such technology, ABE-mediated correction of the c.1824C > T HGPS point mutation in mice was recently found to rescue the vascular phenotype and double the life-span[25]. However, in vivo treatment of pathogenic mutations by base editing likely results in heterogeneous edited tissues[25,26], and the effects of a partially-edited cell population on tissue homeostasis and phenotypical outcome, especially in a highly proliferative tissue such as the skin, have not yet been analysed.

Current studies in mice have raised safety concerns about using adeno-associated viral (AAV) vectors for gene therapy, illustrated by the development of hepatocellular carcinoma, likely caused by integration of the AAV genome in oncogenic hotspots[25,27,28]. Additionally, a long-term study of a haemophilia animal model treated with an AAV that expresses the canine factor VIII, revealed that AAV integration near genes involved in cell growth were clonally propagated[29]. Hence, gene editing approaches mediated by transient non-integrative delivery systems should be further investigated to avoid possible negative effects from an integrative delivery system.

Here we report correction of the most common HGPS mutation, which affects more than 90% of the patients (*LMNA* c.1824C > T) by ABE treatment in HGPS patient-derived B-lymphoblasts and in highly regenerative skin tissue of HGPS mice. This inducible humanized HGPS mouse model has one copy of a human *tetop-LA*$^{G608G}$ minigene with the *LMNA* c.1824C > T mutation in exon 11 in every cell[30–32]. Detailed characterization of the humanized HGPS mouse model has shown that postnatal expression of progerin, under the regulation of the *Krt5* promotor, results in a progressive skin phenotype which is characterized by epidermal hypoplasia, resting hair follicles, associated hypoplastic sebaceous glands, loss of hypodermis and a fibrotic dermis, similar to what is seen in HGPS patients and suggesting that this is a useful model for studying HGPS[33]. For in vivo ABE delivery we used MS2 bacteriophage-lentivirus chimeric particles (LentiFlash®). This vector system transiently expresses the ABE and single-guide RNA (sgRNA) that are packaged in one viral particle as mRNA, and has a transgene expression peak of 4–5 h post-injection. Furthermore, the mRNA molecule delivered by the LentiFlash® system is directly expressed in the cytoplasm and does not contain any retroviral sequences therefore no host genome integration of the viral transgene can be detected[34]. ABE treated skin of HGPS mice at three weeks of age (P21) showed a mutation correction of 20.8–24.1%, which could be detected two days after the initial LF-ABE treatment period. Four weeks post-ABE treatment, injected skin samples demonstrated an improved HGPS phenotype with a reduction of progerin-expressing cells and a formation of progerin negative cell clusters in the basal and suprabasal layers of the epidermis, indicating mutation correction of skin progenitor cells.

## Results

**ABE-mediated correction of the *LMNA* c.1824C > T mutation in patient cells**. To correct the *LMNA* c.1824C > T mutation, we designed a sgRNA with the c.1824C > T HGPS mutation at the protospacer position A6 in reference to the protospacer adjacent motif (PAM). In combination with the sgRNA, we used an ABEmax-VQR, adenine base editor (ABE) that recognizes a 5'-NGA-3' PAM variant (Fig. 1a). To investigate the ABE editing efficiency of the c.1824C > T point mutation, we transfected HGPS patient-derived B-lymphoblasts with ABE and sgRNA-expressing plasmids at a mass ratio of 1:1 and 3:1 (Supplementary Fig. 1). Transfected HGPS patient cells revealed high base conversion of the c.1824C > T mutation towards the correct c.1824G:C allelic fraction of 87.7–99.5% across all replicates with low unwanted insertions or deletions (indels) ranging from 0.008–0.44% within the protospacer region (*n* = 2 in each group). However, we detected 1.4–7.9% (mean 3.95% and 0.75% at 1:1 and 3:1 ABE to sgRNA plasmid mass ratios respectively) base conversion (A:T to G:C) at the protospacer position 10 (A10) in two of the four edited HGPS cell lines (Fig. 1b). This bystander editing, c.1820T > C, is predicted to cause an amino acid transition (V607A) and in silico analysis suggests that this base transition significantly reduces progerin splicing (Supplementary Table 1). The identified A:T to G:C transition at position A10 is predicted to substantially reduce internal exon 11 splicing, even below the value for the reference sequence, both in the presence or absence of base editing at position A6 (Supplementary Table 1). Next, we identified 42 potential off-target loci by digenome-sequencing (digenome-seq)[35] and 19 potential off target loci with the Cas-offinder algorithm[36]. In vitro validation of the off-target loci identified by digenome-seq. (Fig. 1c, Supplementary Fig. 2, Supplementary Data 1) and with the Cas-offinder algorithm (Supplementary Fig. 3, Supplementary Data 1) did not show an increase in the mutation frequency above background levels. Comparing the c.1824C allele frequency by targeted deep sequencing with a droplet digital PCR rare event detection (ddPCR-RED) assay for the c.1824C > T mutation we observed similar allele fractions. By targeted deep sequencing we observed an increase of the average c.1824G:C allelic fraction from 50.5% (in accordance with a heterozygous mutation) up to 92.5–95.8% (Fig. 1d). With the c.1824C > T specific ddPCR-RED assay we measured an increase of the average c.1824C allelic fraction from 49.8% up to 92.4–92.6% (Fig. 1e). This indicated, that both methods are feasible to analyze ABE editing efficiency.

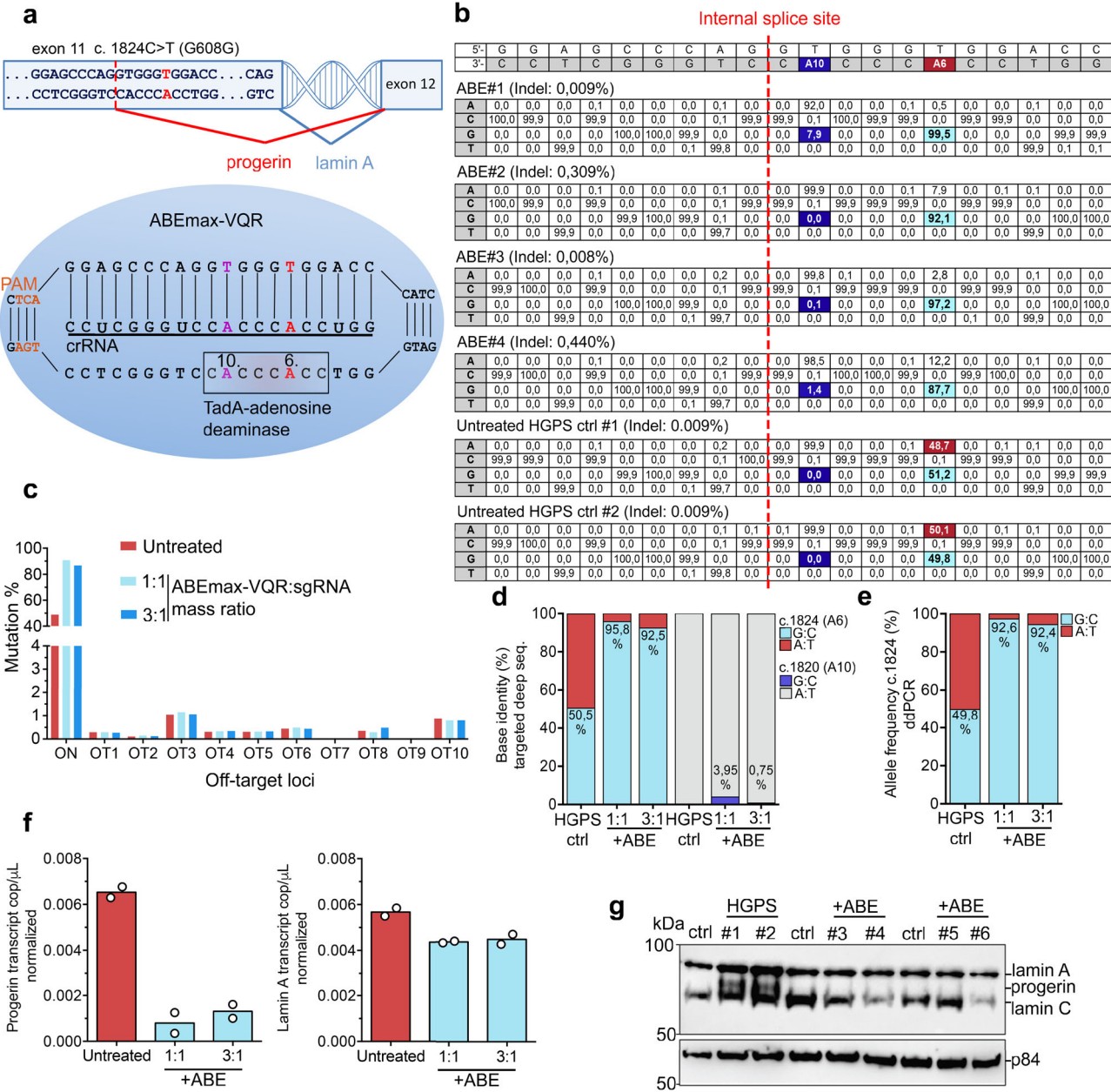

**Fig. 1 ABE-mediated correction of the pathogenic HGPS mutation in patient cells. a** The HGPS-causing point mutation c.1824C > T (p. G608G) was corrected by ABEmax-VQR that recognizes the PAM variant 5′-NGA-3′ (orange). The crRNA part of the sgRNA is positioning the c.1824C > T mutation at the protospacer position A6 (red) in reference to the PAM sequence. **b** Targeted deep-seq. of the protospacer region shows high mutation (A6, red) editing and low bystander (A10, purple) editing in transfected HGPS patient B-lymphoblasts (#1–2, ABE to sgRNA ratio 1:1, #3-4 ABE to sgRNA ratio 3:1, untreated HGPS ctrl #1-#2). **c** Targeted deep-seq. validation of potential off-target loci shows no increase in off-target mutation frequency in comparison to untreated controls (shown; 10 off-target loci, full analysis in Supplementary Fig. 2c & Fig. 3b, n = 2 independent experiments). **d, e** Comparison of the c.1824C allelic fraction as measured by targeted deep-seq. analysis and by a *LMNA* c.1824C > T ddPCR assay (red, c.1824 A:T; blue, c.1824 G:C; grey, c.1820 A:T; purple, 1820G:C). **f** Progerin and lamin A transcripts analyzed by ddPCR-absolute quantification (ddPCR-ABS), untreated samples are depicted in red, ABE treated samples are depicted in blue. **g** Detection of progerin levels by western blot in ABE-transfected HGPS patient B-lymphoblasts (+ABE; #3, #4, ABEmax-VQR: sgRNA mass ratio 1:1 and #5, #6 ABEmax-VQR: sgRNA mass ratio 3:1), untreated HGPS patient B-lymphoblasts (HGPS #1 and #2) and unaffected sibling control B-lymphoblasts (ctrl) repeated with n = 2 independent experiments with similar results. Data are presented as mean values. Source data for (**d–g**) are provided as a Source Data file.

However, low levels of unwanted indels and bystander editing were observed by targeted deep sequencing of the protospacer region. Next, we analyzed progerin and lamin A transcript levels and detected both transcripts in edited HGPS patient cells (Fig. 1f). By progerin protein analysis we could not detect progerin levels in edited HGPS patient cells (Fig. 1g, Supplementary Fig. 2). Overall, ABE mediated mutation correction of HGPS

patient derived B-lymphoblasts resulted in a high mutation correction without introducing off target mutations. Furthermore, corrected HGPS patient cells displayed a reduced expression of progerin mRNA and protein.

**Base editing of the HGPS mutation in mice by transient ABE delivery.** Encouraged by the degree of mutation correction

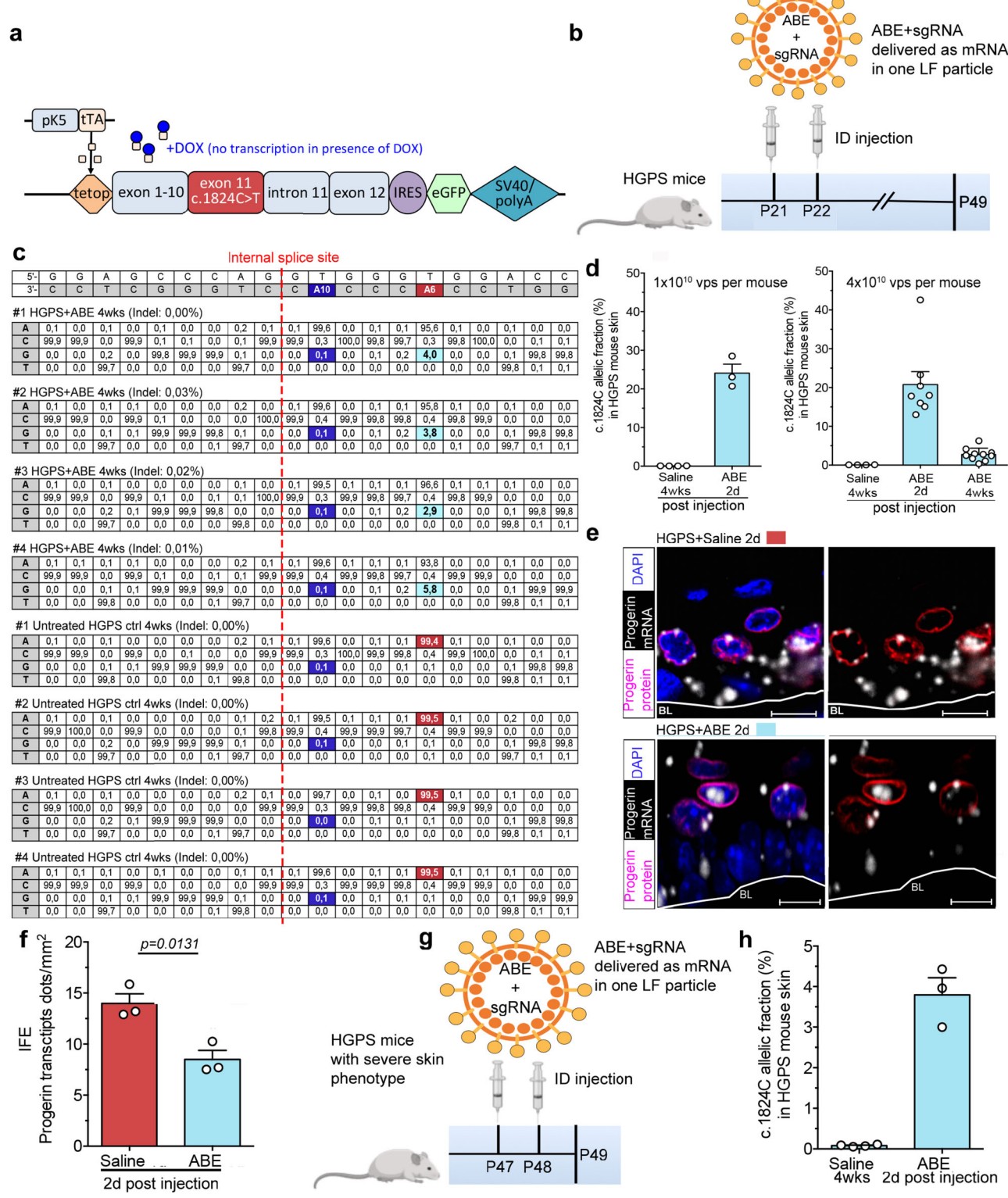

in vitro we decided to test the ABE in a highly proliferative tissue such as skin. Previous studies have provided detailed characterization of the skin phenotype in an inducible humanized HGPS mouse model[33], and given that skin is one of the first tissues to present a phenotype in HGPS patients, we decided to test the ABEmax-VQR in this HGPS mouse model. These HGPS mice have one copy of an inducible human tetop-LA$^{G608G}$ minigene with the LMNA c.1824C > T mutation regulated by the Krt5 promoter, expressing both human progerin and lamin A in

keratinocytes of basal epithelia and peripheral cells of the sebaceous glands[33] (Fig. 2a). First, the ABE and sgRNA were delivered by intraperitoneal injection (IP) using a trans-splicing AAV5 vector system at a dosage of $10^{11}$GC per mouse[37]. However, the LMNA c.1824T mutation correction efficiency by trans-splicing AAV5 delivery of the ABE was less than 0.5% in multiple tissues including the skin (Supplementary Fig. 4). Since recent safety concerns about the usage of AAV vectors were emerging, we decided to test a transient non-integrative base editing

**Fig. 2 In vivo base editing of HGPS mouse skin by transient non-integrative lentiviral vector delivery. a** Description of the human tetop-LA[G608G+] minigene containing the common c.1824C > T point mutation in exon 11 of *LMNA*. In the presence of doxycycline (DOX) the transcription is inhibited. Bi-transgenic animals K5tTA[+], tetop-LA[G608G+] express the HGPS c.1824C > T mutation[30]. **b** Base editing of the c.1824T point mutation in humanized HGPS mice with keratinocyte-specific expression of progerin. HGPS mice were ID injected with either LF-ABE or saline solution at the age of 21 days (P21) for two consecutive days. Treated skin was then analyzed two days (2d) or four weeks (4wks) post-LF-ABE injection. **c** Targeted deep-seq. analysis of the protospacer region shows c.1824 A:T to G:C mutation correction ranging from 2.9% to 5.8% and no bystander editing in HGPS mouse skin (#1–#4, $n = 4$) at 4wks post-injection with $4 \times 10^{10}$ total LF-ABE vps. We could detect a very low formation of indels within the protospacer region in ABE treated HGPS skin. **d** Frequencies of the c.1824C allele fraction in HGPS mouse skin obtained by ddPCR-RED 2d after LF-ABE treatment with either $1 \times 10^{10}$ or $4 \times 10^{10}$ total vps ($n = 3$ and $n = 8$, biologically independent samples respectively), or 4 wks post LF-ABE ($4 \times 10^{10}$ total vps, $n = 10$ biologically independent samples) or saline injection ($n = 4$ biologically independent samples). **e, f** Progerin transcript quantification by in situ hybridization two days post-injection with $4 \times 10^{10}$ total vps of the LF-ABE ($n = 3$ biologically independent samples) shows a significant reduction of progerin transcripts across the IFE ($p = 0.0131$). The first cell layer on top of the white line is defined as the basal skin layer (BL). **g, h** LF-ABE treatment of HGPS mice with a developed skin phenotype at the age of 47 days (P47), showed a c.1824T mutation correction frequency of 3.8% ($n = 3$ biologically independent samples) two days post-injection. Scale bar: d = 10 µm. Saline-treated samples are depicted in red, LF-ABE treated samples are depicted in blue. For (**d, f, h**) data are presented as mean values +/− SEM. **f** $P$ value was calculated by two-tailed unpaired $t$ test, 95% CI. Source data are provided as a Source Data file for (**d, f, h**).

approach with a fast transgene expression peak, five hours post-injection in vivo. Here we used a chimeric MS2-lentiviral particle vector system called LentiFlash® (LF). With the LF vector system the ABEmax-VQR and the *LMNA*[G608G] targeting sgRNA are expressed as mRNA[34], avoiding the trans-splicing step and the risk of random integration of the transgene into the host genome. We treated HGPS mice by intradermal administration (ID) of LF-ABE viral particles (vps) at three weeks of age (P21) with a total dosage of $1 \times 10^{10}$ or $4 \times 10^{10}$ vps divided into a two-day treatment period (Fig. 2b). Two days after the initial LF-ABE treatment of HGPS mice, we observed an average editing frequency of 24.1% ($n = 3$, ranging from 20.7–28.5% in individual samples) and 20.8% ($n = 8$, ranging from 13% to 43.8% in individual samples) of the skin cells, at the lower and higher doses respectively measured by ddPCR-RED (Fig. 2d). The measured allelic fraction likely reflects an intermediate editing stage where the adenine (A) is converted to an inosine (I) where cell replication and DNA repair mechanisms have not yet incorporated the final guanine (G), cytosine (C) base pair[24]. However, this intermediate editing stage was not detected to the same extent by targeted deep sequencing after conventional PCR amplification, indicating a reduced polymerase amplification efficiency in the presence of inosine nucleotides[38,39] (Supplementary Fig. 5). Skin is normally fully regenerated after 8–9.5 days in mice[40], and in order to examine if a remaining edited cell population was still present after the skin had regenerated we assessed LF-ABE injected HGPS mouse skin four weeks post-injection with $4 \times 10^{10}$ total LF-ABE vps. As measured by ddPCR-RED, analysis of ABE-LF injected HGPS skin demonstrated a remaining corrected cell fraction of 2.7% ($n = 10$) which we confirmed by targeted deep sequencing of the protospacer region, where we detected a remaining corrected cell fraction of 4.1% (average of #1–4, $n = 4$) (Fig. 2c, d). The basal layer of the interfollicular epidermis (IFE) contains skin progenitor cells that divide and form new progenitor and differentiated cells, that migrate outwards and are eventually lost on the surface epithelium within one week due to the quick regenerative capacity of the skin[40,41]. Given that a remaining fraction of cells that retained the *LMNA* c.1824C WT allele are still present within the tissue after four weeks, our observation suggests that mostly differentiated cells were edited at the initial LF-ABE injection but also a fraction of progenitor cells which are still present in the tissue even after several regeneration cycles of the skin. Furthermore, assessment of unwanted bystander (A10) editing within the protospacer region after four weeks post-LF-ABE treatment ($4 \times 10^{10}$ total vps) did not indicate such events in HGPS skin samples ($n = 4$) (Fig. 2c) Similarly, unwanted indel formation within the protospacer region was observed at a very low frequency (0.0–0.03%). Since we could detect a fast HGPS

mutation correction already two days after the initial injection of $4 \times 10^{10}$ total LF-ABE vps, we assessed progerin transcripts by in situ hybridization. Here, we detected a significant reduction in progerin transcripts in the interfollicular epidermis (IFE) of LF-ABE treated HGPS mice ($n = 3$, $p = 0.0131$) confirming fast mutation correction by LF-ABE delivery and downregulation of the cryptic splice event (Fig. 2e, f). We also assessed the LF-ABE efficiency in HGPS mice with a more severe skin phenotype at the age of seven weeks (P47-P48). However, ID delivery of the LF-ABE ($4 \times 10^{10}$ total vps) resulted in a fraction of 3.8% of corrected skin cells, two days post-injection (Fig. 2g, h). This result suggests a lowered ABE correction and/or viral particle delivery efficiency in more severely affected HGPS skin.

**Transient ABE delivery improves the HGPS skin phenotype.** Next, we analysed whether the LF-ABE-treatment could result in long-term improvements of the skin. Here, we focused on mice that received the LF-ABE ($4 \times 10^{10}$ total vps) at P21-22 and analyzed the skin phenotype four weeks after the LF-ABE treatment period when the mice where at 7 weeks of age. At this age, saline-treated as well as untreated HGPS mice have developed a severe skin phenotype with regional variations, and showed epidermal hyperplasia and hyperkeratosis, hyperplasia of sebaceous glands, apoptosis of sebocytes, and an increase in inflammation[33]. LF-ABE-treated HGPS skin tissues collected four weeks post-injection displayed a less severe skin phenotype in comparison to uninjected adjacent skin (next to the LF-ABE injection site) and saline-treated HGPS skin (Fig. 3a–d). Histopathological analysis revealed regional variation of the epidermal thickness (as previously described for this model[30]). A significant reduction of the average epidermal thickness of the interfollicular epidermis was observed after LF-ABE-treatment, from 38.6 µm in uninjected adjacent skin regions ($n = 4$) and 37.7 µm in saline-treated HGPS skin to 22.6 µm in LF-ABE-treated HGPS skin ($n = 6$, $p = 0.0021$ & $p = 0.0020$ respectively), (Fig. 3e). Furthermore, we observed a reduction of inflammation in LF-ABE-treated HGPS mice ($n = 7$, $p = 0.0277$) in comparison to adjacent uninjected skin regions (Fig. 3f). Next, we assessed whether progerin protein and transcript levels were reduced four weeks after the initial LF-ABE injection. We observed a significant reduction in progerin transcript levels in LF-ABE treated HGPS mice ($n = 4$, $p < 0.001$). These changes in progerin transcript did not affect the lamin A transcript abundance (Fig. 3g), in agreement with lamin A transcript levels being unaffected by progerin splicing[42]. By performing additional protein analysis with protein extracts from the whole skin, we could not observe a significant difference in progerin protein abundance in LF-ABE-treated skin samples

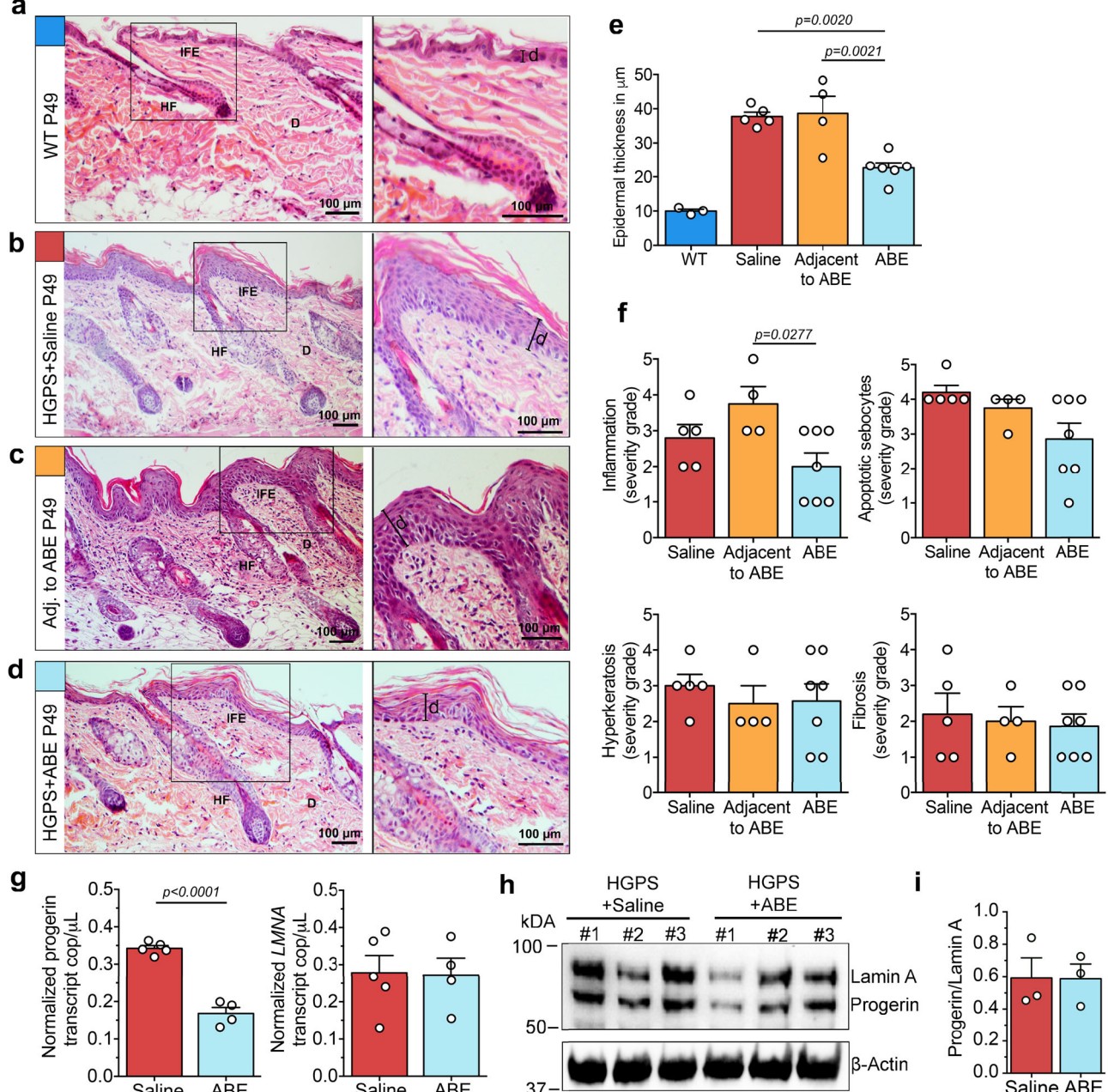

**Fig. 3 Long-term improvements of HGPS mouse skin pathology after LF-ABE treatment. a–d** Skin from wild-type mice (WT, $n = 3$ independent experiments with similar results) and saline ($n = 5$, independent experiments with similar results), uninjected adjacent skin sites ($n = 4$, independent experiments with similar results) or LF-ABE ($n = 6$, independent experiments with similar results) treated HGPS mice was collected at seven weeks of age (P49) and H&E-stained for histopathological analysis (Interfollicular epidermis, IFE; hair follicle, HF; dermis, D, scale bar: $d = 100 \, \mu m$). **e** Graph showing the significant reduction of the epidermal thickness (indicated as **d**) in LF-ABE treated ($n = 6$ biologically independent sample) vs. saline-treated HGPS ($n = 5$ biologically independent samples, $p = 0.0020$) and uninjected adjacent skin sites ($n = 4$ biologically independent samples, $p = 0.0021$). **f** Graphs showing the histopathological assessment by severity grading (1 = minimal, 2 = mild, 3 = moderate, 4 = marked, 5 = severe) of inflammation (Adjacent to ABE compared to ABE, $p = 0.0277$), apoptotic sebocytes, hyperkeratosis and fibrosis in LF-ABE ($n = 7$ biologically independent samples), saline-treated HGPS mice ($n = 5$ biologically independent samples) and uninjected adjacent skin sites ($n = 4$ biologically independent samples). **g** Absolute quantification (ddPCR-ABS) was used to measure progerin (Saline compared to ABE, $p < 0.0001$) and lamin A transcript levels in LF-ABE ($n = 4$ biologically independent samples) and saline-treated HGPS mouse skin ($n = 5$ biologically independent samples) four weeks post-treatment. **h, i** Western blot analysis of progerin abundance ($n = 3$ biologically independent samples, #1–3 for each LF-ABE treated and saline treated group). WT samples are depicted in blue, uninjected adjacent skin sites are depicted in yellow, saline-treated samples are depicted in red LF-ABE treated samples are depicted in light blue. Data are represented as mean values $+/-$ SEM. **e**, **f** $P$ values were calculated by one-way ANOVA, adjusted for multiple comparisons, 95% CI. **g** $P$ value was calculated by two-tailed unpaired $t$ test, 95% CI. For (**e–i**) data are provided as a Source Data file.

(Fig. 3h, i). Taken together, LF-ABE treated mouse skin displayed an improved phenotype four weeks post-treatment which indicates a steady effect from the otherwise transient delivery system.

**LF-ABE treated skin displays clusters of progerin negative epidermal cells.** In order to investigate the previously described improvements of the HGPS phenotype on a cellular level and relate them to progerin expression, we performed progerin immunofluorescence staining and progerin in situ hybridization to assess progerin protein and transcript levels in the same section. Random region assessment of progerin expression by immunofluorescence staining of LF-ABE-treated HGPS skin mice revealed a significant reduction of 10.2% in the total frequency of progerin-expressing cells in the basal layer ($n = 5$, $p = 0.0422$). However, levels of progerin-expressing cells in the IFE and suprabasal layer of the skin were unchanged (Fig. 4a, b). We then examined the IFE of LF-ABE-treated mice closer and detected formations of large clusters of up to 81 progerin negative per K5 positive cells (Supplementary Fig. 6). Analysis of these progerin-negative per K5 positive cell clusters revealed an average cluster size of 10.6 cells across the whole skin section, four weeks after the initial LF-ABE-treatment ($n = 5$, $p = 0.0023$). Saline-treated controls displayed an average cluster size of 1.6 progerin negative per K5 positive cells/cluster with no clusters larger than five cells in the IFE (Fig. 4c, Supplementary Fig. 6). In addition we analysed progerin transcript levels by in situ hybridization in LF-ABE and saline-treated HGPS mice (Fig. 4d, e, Supplementary Fig. 7), and detected a significant reduction of progerin transcripts in cell clusters that lacked progerin protein expression ($n = 3$, $p = 0.0082$). Next, we assessed the accumulation of progerin, in the nuclear lamina by measurement of the fluorescence intensity. We observed a significant reduction of progerin, after normalization to the DNA content per cell across the IFE ($n = 3$, $p = 0.0098$) and specifically in both the basal ($n = 3$, $p = 0.0141$), and suprabasal layer ($n = 3$, $p = 0.0459$) of LF-ABE-treated mice (Fig. 4f). This observation indicates a lower degree of progerin accumulation in HGPS skin after treatment with the LF-ABE, likely resulting from the improved tissue homeostasis and from the reduced epidermal hyperplasia caused by the clusters of progerin-negative cells that contribute to the tissue normalization. Taken together, our results suggest that the LF-ABE treatment reduced the frequency of progerin-expressing cells, both differentiated cells in the suprabasal layer and progenitor cells, positioned in the basal layer. Furthermore, we detected large cell clusters in the IFE that lacked progerin protein and transcript expression, suggesting an expansion of mutation corrected epidermal progenitor cells that contributed to the improvement of tissue homeostasis following tissue regeneration.

**Transient ABE expression decreases the amount of DNA damage and increases keratin 15 expression in progerin negative cell clusters.** Since we could observe a reduction in progerin protein and transcripts, we next examined a possible improvement in the cellular phenotypes evident upon progerin expression. Previous studies have shown that expression of progerin results in the activation of the DNA damage response pathway[43,44]. LF-ABE treated HGPS mice, showed a significantly reduced accumulation of the DNA damage marker 53BP1 in the IFE ($n = 3$, $p = 0.0139$) and specifically in the basal cell layer of progerin-negative cell clusters ($n = 3$, $p = 0.0314$) compared to saline-treated controls (Fig. 5a, b). Previous studies using this HGPS mouse model have shown that expression of progerin results in a reduction of cells with stem and progenitor characteristics and that the impaired stem cell function is to some extent reversible if progerin expression is turned off[31,33]. Keratin-

15 has been suggested as an epidermal stem cell marker for basal keratinocytes and has been shown to be downregulated upon terminal differentiation and cell migration towards the surface epithelium[45,46]. Earlier work conducted on the same model has shown that *Krt15* is downregulated upon progerin expression in basal keratinocytes[47]. To analyze the expression of *Krt15*, we performed in situ hybridization and detected an increase in *Krt15* transcripts ($n = 3$, $p = 0.0406$) in the basal layer of LF-ABE treated mice. This increase was restricted to cell clusters that lacked progerin expression, four weeks after the initial LF-ABE-treatment of HGPS mice (Fig. 5c, d). Furthermore, when we assessed *Krt15* transcript abundance by ddPCR in RNA extracted from LF-ABE treated skin we detected a trend of increased *Krt15* transcripts (Fig. 5e). Similar results were seen using immunofluorescence with antibodies for keratin-15 and progerin proteins in progerin-negative cell cluster regions of the basal epidermis (Fig. 5f, g). Taken together our findings suggest that HGPS mutation correction by LF-ABE treatment reduced the accumulation of DNA damage by reducing progerin expression levels and further increased the expression of *Krt15*, improving the tissue homeostasis of the skin.

## Discussion

Here we report ABE correction of the common c.1824C > T HGPS mutation of HGPS patient cells but also in vivo by transient non-integrative lentiviral ABE delivery in HGPS mice. We observed that the LF-ABE treatment in HGPS mice is initially correcting an average of 24.1% and 20.8% of the skin cells, with a fraction of corrected cells ranging from 13% to 43.8% between individual samples. Since ABEs convert adenine (A), thymine (T) base pairs to inosine (I), cytosine (C) base pairs with a nick on the opposing unedited strand, the final guanine (G), cytosine (C) editing outcome is dependent on replication and DNA repair[24]. Hence, the observed corrected cell fraction of 24.1% and 20.8% two days post LF-ABE treatment likely in part reflects an intermediate editing state where the inosine (I) is still present and can be amplified by the ddPCR-RED assay in comparison to targeted deep sequencing that provided a lower fraction of edited cells. Further, the nick of the opposing unedited strand is likely diminishing progerin splicing, suggested by the observed reduction of progerin transcripts, two days post LF-ABE treatment of HGPS mice. Recently, Koblan et al., 2021 detected a larger reduction in progerin transcripts in comparison to the fraction of corrected alleles, using an ABE to correct the HGPS mutation in mice. In a tissue with low regeneration and a slower cell replication and DNA damage repair the intermediate editing stage is likely prolonged and the frequency of corrected alleles can therefore be higher than detected by sequencing methods, suggesting that ddPCR-RED analysis would provide a more accurate read-out.

In addition, we observed bystander editing at the A10 position (*LMNA* c.1820 A:T to G:C) in HGPS patient-derived B-Lymphocytes. In silico splice site prediction revealed that editing of this position significantly reduced the splice score, even substantially below the reference sequence. This suggests that editing at the A10 position (either by itself or together with the correction at the A6 position) would have a significant impact on reducing progerin splicing.

Four weeks after the delivery of the LF-ABE we observed a remaining edited cell population and an improved skin phenotype, including clusters of cells in the IFE where progerin was absent. In addition, we observed an improved tissue homeostasis, confirmed by the less pronounced epidermal hyperplasia likely caused by the reduced time for the cells to remain in the tissue to accumulate progerin before being replaced by new cells that migrate out from the basal layer towards the surface epithelium.

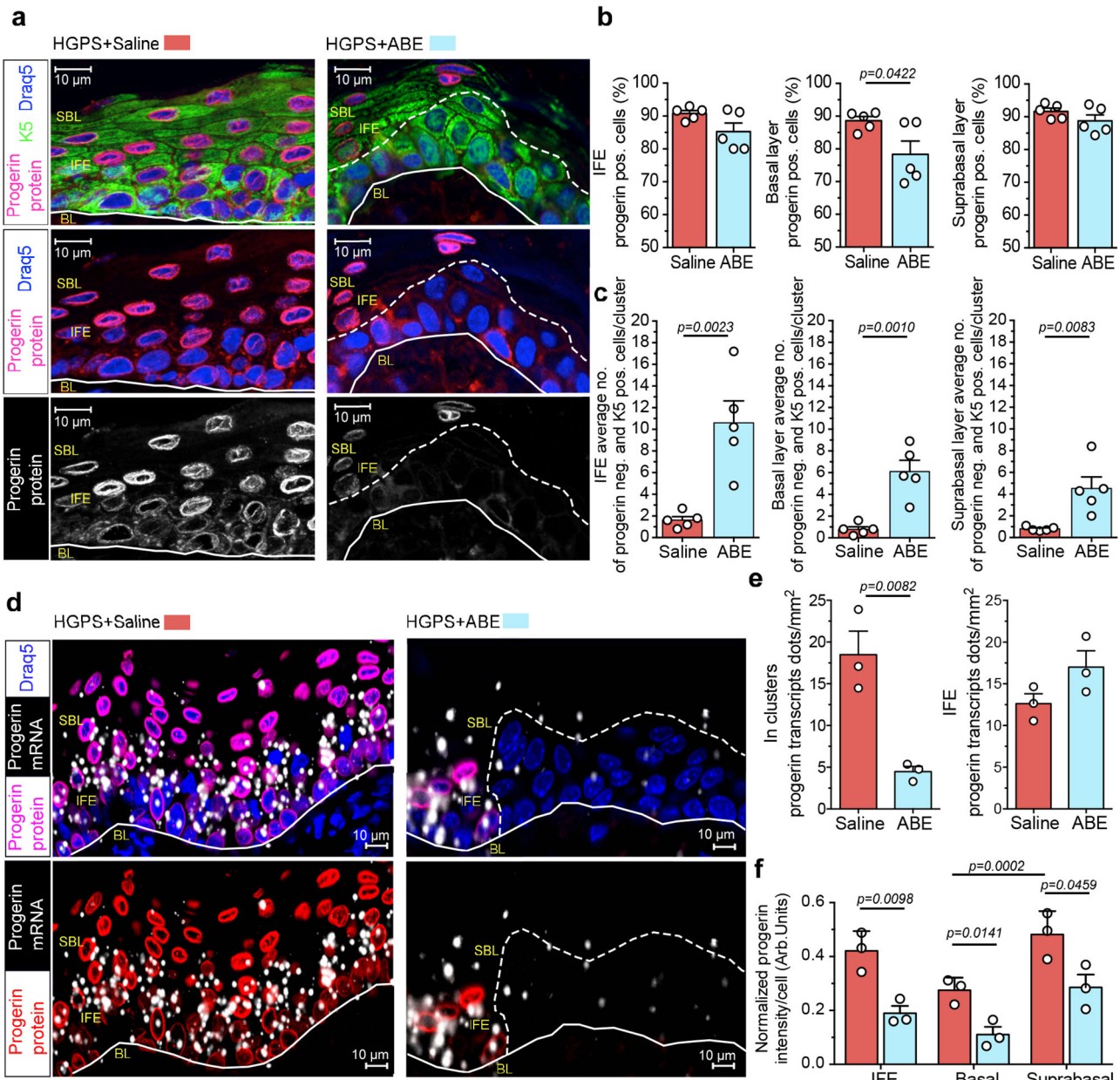

**Fig. 4 LF-ABE treatment of HGPS mouse skin reduces the amount of progerin-expressing cells in the basal progenitor layer of the skin. a** Progerin and keratin-5 (K5) immunofluorescence staining of LF-ABE and saline-treated HGPS skin collected four weeks after treatment. **b** Quantification of the frequency of progerin protein expressing cells in the IFE, and specifically in the basal (Saline compared to ABE, $p = 0.0422$) and suprabasal layers of LF-ABE ($n = 5$ biologically independent samples) and saline-treated ($n = 5$ biologically independent samples) HGPS skin. **c** The number of progerin negative per keratin-5 positive cells/cluster in the IFE (Saline compared ABE, $p = 0.0023$), and specifically in the basal (Saline compared ABE, $p = 0.0010$) and suprabasal layers (Saline compared ABE, $p = 0.0083$) of LF-ABE ($n = 5$ biologically independent samples) and saline-treated ($n = 5$ biologically independent samples) HGPS skin is shown. **d**, **e** Progerin transcript levels measured by progerin in situ hybridization (**d**) in the IFE and in clusters (Saline compared to ABE, $p = 0.0082$) of progerin negative cells (**e**) in LF-ABE treated skin ($n = 3$ biologically independent samples) in comparison to saline-treated HGPS skin ($n = 3$ biologically independent samples). **f** Decreased progerin accumulation in the IFE ($p = 0.0098$) and in the basal ($p = 0.00141$) and suprabasal skin layers ($p = 0.00459$) four weeks after LF-ABE ($n = 3$ biologically independent samples) treatment in comparison to saline-treated HGPS skin ($n = 3$ biologically independent samples) measured by progerin intensity normalized to DAPI intensity (Arbitrary Units = Arb. Units) Significant progerin accumulation ($p = 0.002$) between the basal ($n = 3$ biologically independent samples) and suprabasal skin layer ($n = 3$ biologically independent samples) of saline-treated HGPS mice. BL = basal skin layer, the first cell layer on top of the white line; cluster of progerin negative cells are delineated with a dotted white line, SBL = suprabasal skin layer, IFE = interfollicular epidermis. Scale bars: a, d d = 10 μm. Saline-treated samples are depicted in red, LF-ABE treated samples are depicted in blue. Data are represented as mean values +/− SEM. **b**, **c**, **e**, **f** $P$ values between samples were calculated by two-tailed unpaired $t$ test, 95% CI. **f** $P$ value between groups was calculated by one-way ANOVA, adjusted for multiple comparisons, 95% CI. For (**b**, **c**, **e**, **f**) data are provided as a Source Data file.

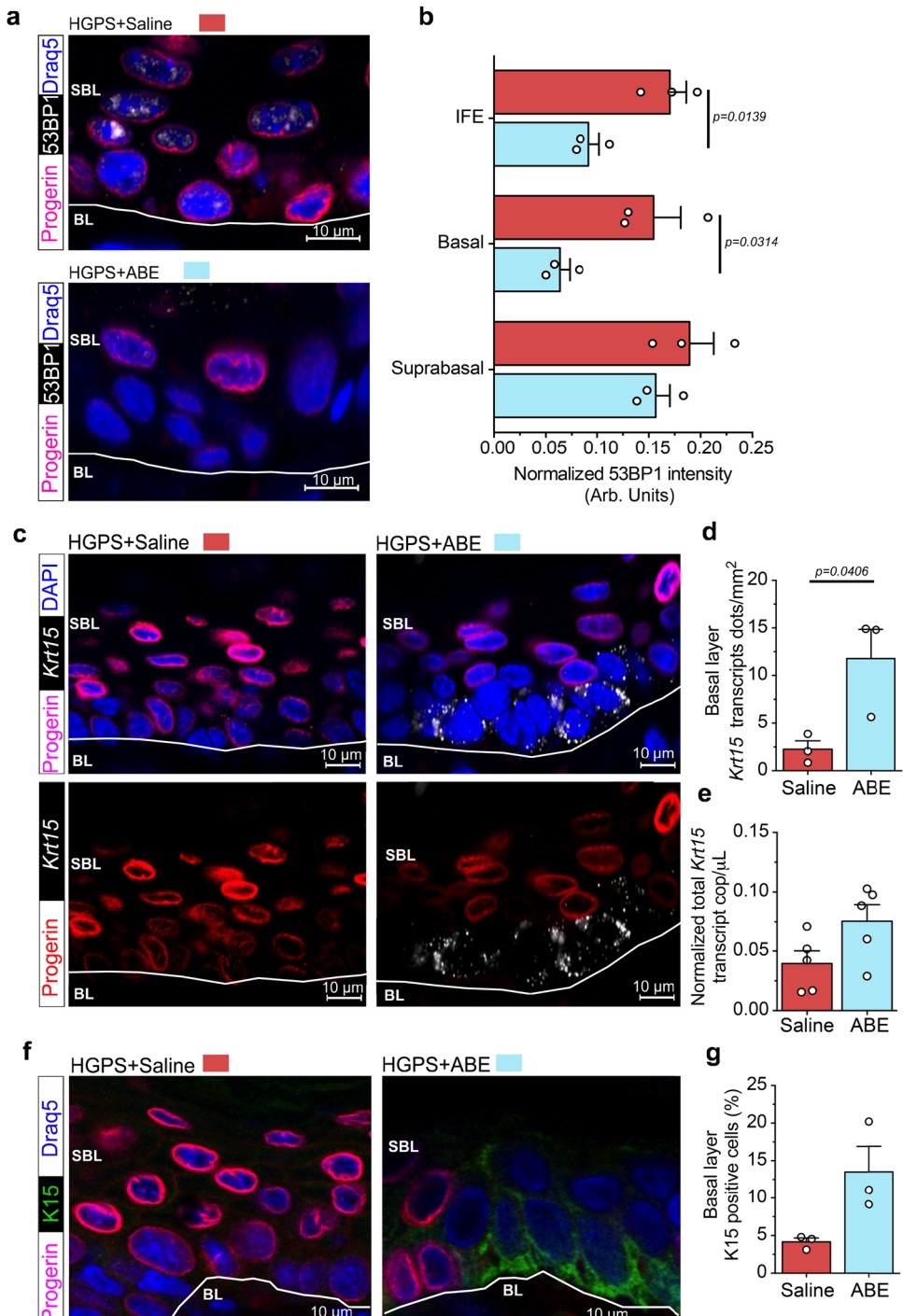

**Fig. 5 Reduced accumulation of DNA damage and increased expression of Krt15 in clusters of cells without progerin expression. a** Colocalization staining of progerin and 53BP1 in LF-ABE and saline-treated HGPS mouse skin four weeks post-injection. **b** In the IFE and the basal layer of LF-ABE treated skin (*n* = 3 biologically independent samples) in comparison to saline-treated controls (*n* = 3 biologically independent samples), a reduction in 53BP1 intensity was observed in progerin negative clusters, in the IFE (*p* = 0.0139) and basal layer (*p* = 0.0314) along with a reductive trend of 53BP1 accumulation in the suprabasal layer of the skin (53BP1 intensity normalized to DAPI intensity, Arbitrary Units = Arb. Units). **c** *Krt15* in situ hybridization colocalized with progerin protein staining in LF-ABE and saline-treated HGPS mouse skin. **d** Increased *Krt15* transcripts (*p* = 0.0104) in skin upon LF-ABE treatment (*n* = 3 biologically independent samples) in comparison to saline-treated controls (*n* = 3 biologically independent samples) four weeks post-injection. **e** Absolute quantification of Krt15 transcripts by ddPCR in LF-ABE (*n* = 5 biologically independent samples) and (*n* = 5 biologically independent samples) saline-treated mouse skin. The first cell layer above the white line is defined as basal layer (BL). **f, g** Colocalization staining of progerin and keratin-15 (K15) proteins in LF-ABE (*n* = 3 biologically independent samples) and saline-treated (*n* = 3 biologically independent samples) HGPS mouse skin four weeks post-injection with quantification graph. Scale bars: a, c, d = 10 μm. LF-ABE treated samples are depicted in blue, saline-treated samples are depicted in red. Data are represented as mean values +/− SEM. **b, d** *P* values between samples were calculated by two-tailed unpaired *t* test, 95% CI. For (**b, d, e, g**) data are provided as a Source Data file.

Epidermal skin homeostasis is dependent on the symmetry of cell division of epidermal progenitors located in the basal layer, and on the migration of keratinocytes through the suprabasal layer of the epidermis. If the skin homeostasis is not balanced between regeneration of new cells and shedding off of old cells, then the epidermis may become hyperplastic, as is the case in the presently used mouse model[33,47]. Even though the skin differs between mice and humans, the end-stage of HGPS mouse skin shares phenotypic features with HGPS patients skin, which makes this mouse model a useful tool for the disease[33]. Thus, the long-term improvements seen in the tissue, four weeks after the initial mutation correction, suggest that to some extent the skin tissue homeostasis has been restored, likely as a consequence of the editing of a progenitor population and the reduced accumulation of progerin per cell. Indeed, the remaining fraction of corrected cells may reflect the editing of a progenitor cell population, especially since it has been shown that mice epidermis is fully regenerated after one week[40,41]. In accordance, immunofluorescence staining of progerin demonstrated a reduction of progerin-expressing cells in the basal layer of the skin where skin progenitors cells reside[41,48–50]. Furthermore, we observed the formation of cell clusters without progerin expression in the IFE of LF-ABE treated mice, where the basal skin layer mostly contributed to the formation of these progerin negative cell clusters.

Interestingly, HGPS mouse skin treated at seven weeks of age, with a progressed HGPS phenotype, showed a low HGPS mutation correction of 3.8%. This indicates that a more severe HGPS phenotype may interfere with the base editing and/or viral vector transduction efficiency. Our results suggest that sustained effects from the HGPS mutation correction slows down the accumulation of progerin over time and that the remaining clusters of edited cells help to keep the tissue more accessible to additional rounds of LF-ABE treatment. Thus, an extended LF-ABE treatment period with multiple injections could further improve the HGPS skin phenotype.

Taken together our results suggest that heterogeneous editing of a highly proliferative tissue, such as the skin, by transient non-integrative vector mediated ABE expression can improve the HGPS skin phenotype. The observed phenotype improvements after four weeks post LF-ABE treatment are likely mediated by a corrected progenitor cell population that still reside in the tissue even after the skin is fully regenerated. In addition, in silico splice score prediction of bystander editing at position A10 (LMNA c.1820 A:T to G:C) revealed that this nucleotide position could be a novel target for future base editing approaches in HGPS, either in combination with the A6 position (LMNA c.1824 A:T to G:C) correction or potentially as a solitary target, to significantly reduce progerin splicing. Future base editing approaches for genetic diseases caused by alternative splicing could potentially target key nucleotides of the splice consensus motif to abolish disease-associated splice variations, instead or in combination with the disease-causing mutations. Since transient non-integrative delivery systems and base editors are very versatile, this treatment approach could be used for many genetic diseases, with the possibility of multiple rounds of treatment.

## Methods

**Plasmid construction**. pCMV-ABEmax-VQR-P2A-GFP, a plasmid for mammalian cell expression, and pET-ABE7.10-VQR, a plasmid for bacterial protein expression with an N-terminal His purification tag, were generated by incorporating mutations in the PAM-interacting domain of Cas9 (D1135V, R1335Q, T1337R) into the pCMV-ABEmax-P2A-GFP (Addgene, #112101) and pET-ABE7.10[36]. The sgRNA-encoding plasmid with the 20 nucleotide targeting crRNA sequence: 5′-GGUCCACCCACCUGGGCUCC-3′ was constructed from pRG2 (Addgene, #104174) digested with BsaI.

**WT and HGPS patient B-lymphoblast cell culture**. WT (AG03507) and HGPS patient B-lymphoblast cells (AG03506, Coriell) were maintained with RPMI1640 (ThermoFischer Scientific) supplemented with 15% FBS and 1% penicillin/streptomycin and 2 mM L-Glutamine at 37 °C with 5% carbon dioxide and the cultured cells were passaged two times per week. WT (AG03507) and HGPS patient B-lymphoblast cells (AG03506, Coriell) were authenticated by sequencing[14] and were validated in this study by droplet-digital PCR rare event detection specifically for the LMNA c.1824C > T point mutation.

**Nucleofection of B-lymphoblast cells and fluorescence-activated cell sorting**. B-lymphoblast cells ($1 \times 10^6$) were nucleofected with a total of 5 μg of plasmids encoding ABEmax-VQR-P2A-GFP and sgRNA. Cells were transfected using SF Cell Line 4D-Nucleofector X Kit (Lonza) in a single nucleocuvette (final reaction volume of 100 μL) with FF-113 program according to manufacturer's protocol. At 24 h after transfection, GFP-positive cells were collected using a BD FACSAria III sorter (BD Biosciences) with FACSDiva software (version 6.1.3).

**Splice score analysis**. Splicing was analysed using four different splice score models with the MaxEntScan splice analysing tool[51] (http://hollywood.mit.edu/burgelab/maxent/Xmaxentscan_scoreseq.html).

**Targeted deep sequencing of genomic DNA**. Genomic DNA was isolated from sorted cells using a DNeasy Blood & Tissue Kit (Qiagen) at 72 h after transfection according to manufacturer's protocol. Preparation of sequencing library and targeted deep sequencing was performed as described previously[35]. In brief, the region of interest was first amplified from genomic DNA using KAPA HiFi HotStart DNA polymerase (Roche) to a size of ~500 bp. Then, amplicons were subjected to PCR amplification to label each fragment with index and adapter sequences for TruSeq DNA-RNA CD index system (Illumina). The sequencing libraries were purified using a PCR purification kit (MGmed) and sequenced using a MiniSeq (Illumina) with an average depth of ~5000 reads. Primer sequences used for targeted deep sequencing at on- and off-target sites are listed in Supplementary Data 1. Substitution frequencies at each base position and allelic fractions were measured using BE-analyzer[52]. Indel frequencies were measured using Cas-analyzer[53].

**Digenome-sequencing sample preparation and analysis**. Digenome sequencing was performed as previously described[35]. In brief, genomic DNA isolated from HGPS patient derived B-lymphoblasts was incubated with ABE-VQR protein (extracted from competent E. coli cell transformed with pET-ABE7.10-VQR plasmid) and in vitro-transcribed c.1824C > T-targeting sgRNA, and with Endo V (New England Biolabs). Whole genome DNA libraries were generated by fragmentation, ends repair, 3′ ends adenylation and adapter ligation from a total 1 ug of in vitro ABE- and EndoV-treated genomic DNA using TruSeq Nano DNA Library kit (Illumina). DNA libraries were subjected to whole genome sequencing using a HiSeq X Ten Sequencer (Illumina) at Macrogen. Digenome-positive sites were identified from WGS data by running the Digenome-program.

**Targeted amplicon sequencing (MiSeq.)**. Targeted amplicon sequencing (MiSeq.) was performed at the Bioinformatics and Expression Analysis core facility (BEA) at the Karolinska Institutet. DNA samples from ABE injected skin, two days post LF-ABE treatment were amplified using LMNA forward and reverse primers (LMNART10F/R) with the following PCR cycle conditions: 94 °C, 15 min; 94 °C, 45 s; 57 °C, 30 s; 72 °C, 45 s; 72 °C, 10 min; for 35 cycles. Primers sequences were: LMNART10-Fwd: GCAACAAGTCCAATGAGGACCA and LMNART10-Rev: GTCCCAGATTACATGATGC. After the PCR amplification, DNA samples were column purified using the manufacturers protocol (MinElute PCR Purification Kit, Qiagen) and provided to BEA for sequencing. For amplicon sequencing data analysis the FASTQ files are aligned against reference construct using BWA-mem v0.7.17 algorithm. Once aligned, the PCR duplicate reads are removed from BAM files using Picard v2.10.3 tool MarkDuplicates. Obtained aligned BAM files are sorted and indexed with Samtools v1.12. The SNPs/Mutations are called using Freebayes v1.3.2.

**Animal experiments**. All animal experiments were performed in accordance with the Karolinska institutional guidelines and regulations. All procedures were approved by the Stockholm South Ethical review board (Dnr. 06088-2020). Animals were housed with a 12 h light/dark cycle at 20–22 °C, 50–60% humidity in a pathogen-free animal facility at the Karolinska University Hospital. Bitransgenic HGPS mice were generated by intercrossing heterozygous tetop-LA[G608G] with heterozygous K5-tTA mice with a Fvb/N background to generate LA[G608G]/K5-tTA offspring[33]. In this study, bitransgenic LA[G608G]/K5-tTA and tetop-LA[G608G] mice are referred to as HGPS mice. During LA[G608G]/K5-tTA intercross breedings, mice were supplemented with 100μg/mL Doxycycline in 2.5% Sucrose in the drinking water protected from light and the supplemented water bottles were exchanged every second day. For LF-ABE injections, offspring animals where then injected at P21-22 (ID) with a total virus concentration of $1 \times 10^{10}$ or $4 \times 10^{10}$ total viral particles per mouse divided in two consecutive injections with 50 μL each. For ABE-AAV5 injections, female and male offspring animals were injected with a total

dosage of $10^{11}$GC per mouse divided into two consecutive injections dependent on body weight.

**Intradermal injection and skin collection**. All animals were injected using 0.3 mL × 8 mm (30 G) U-100 insulin syringes (BD, MicroFine) under isoflurane anesthesia. On the first day of the injection period, LF-ABE viral particle aliquot solutions of 50 µL were thawn on ice and loaded into the syringes and injected. On the second day (P22) this process was repeated and the viral particle solution was injected at the exact same site of the previous injection. The injection site was located on the right lower back in all saline and LF-ABE injected animals. All hair was removed by shaving before the injection and the skin area was cleaned with 70% ethanol. After the intradermal injection was performed, the injected skin region was circled with a surgical marker. This procedure was repeated on the following day. All animals were monitored daily and the marking of the injection site was updated twice per week with a removal of regrown hair by shaving. Either 2 days after the first injection or at 4 weeks after the initial injection (p21), animals were sacrificed. In order to exclude dilution and/or variation effects a similar skin tissue size was extracted from all animals. The skin of the marked injection site was extracted in a circular shape (diameter ≈ 1 cm, depth ≈ 1–2 mm) using scissors and forceps and divided. One half of the circular sample was used for DNA, RNA and protein extraction (diameter ≈ 3–5 mm, depth ≈ 1–2 mm), the other half was preserved in 4% paraformaldehyde (PFA) for staining.

**Custom adeno-associated virus serotype 5 (AAV5) production**. Trans-splicing AAV vectors encoding sgRNA and split ABE7.10-VQR (at Gly315-Gln316 of Cas9) were generated using NEBuilder HiFi DNA Assembly Master Mix (New England Biolabs) to insert EF1α promoter and sgRNA sequences into the pAAV-ABE-NT-sgRNA plasmid[26] (addgene #112734) and to incorporate mutations in the Cas9 PAM-interacting domain (D1135V/R1335Q/T1337R) into the pAAV-ABE-CT plasmid[26] (addgene #112876). Mammalian U6-sgRNA and EF1α NT-ABE7.10max-VQR and CT-ABE7.10max-VQR vectors were send to Vigene-Biosience and packaged into AAV5 viral particles (500 mL, $1 × 10^{13}$GC, for each packaging). The custom AAV5 titre was determined by qPCR for the inverted terminal repeats (ITRs) of the viral vector genome by VigeneBiosience and ready to use ABE-NT and ABE-CT AAV5 vector aliquots were injected into HGPS mice.

**Custom LentiFlash® viral particle production**. Mammalian pCMV-ABEmax-VQR-P2A-GFP and *LMNA*G608G targeting sgRNA expression vectors were send to Vectalys by Flash Therapeutics and cloned into an RNA transfer plasmid containing a U6 promotor sequence for the sgRNA and an EF1α promotor sequence for the ABEmax-VQR (RLP.U6.sg1hLMNA.opt.PP7.EF1.ABEmax-VQR.MS2). Titers are obtained by a p24 ELISA directly from the manufacturer Vectalys, using manufacturers protocol for p24 antibody staining. The p24 core antigen is detected directly on the viral supernatant with a HIV-1 p24 ELISA kit provided by Perkin Elmer. The captured antigen is complexed with a biotinylated polyclonal antibody to HIV-1 p24, followed by a streptavidin-HRP (horseradish peroxidase) conjugate. The resulting complex is detected by incubation with ortho-phenylenediamine-HCL (OPD), which produces a yellow color that is directly proportional to the amount of p24 captured. The absorbance of each microplate well is determined using a microplate reader and calibrated against the absorbance of an HIV-1 p24 antigen standard curve. Aliquoted and ready to use LF-ABE viral particles where then injected into mice.

**DNA and RNA collection**. All animals were sacrificed with an excess of isoflurane. Tissues were collected on dry ice for DNA and RNA analysis. DNA isolation was performed according to the manufacturers protocol for tissues or cell pellets (Gentra Purgene, Qiagen). RNA isolation was performed using the TRIzol reagent (Ambion, Life Technologies) manufacturers protocol and RNA extracts were column purified (RNeasy Plus Universal Kit, Qiagen). SuperScript III (Invitrogen, Life Technologies) was used for cDNA synthesis with random hexamers. RNA yield was measured with a NanoDrop ND-100 spectrophotometer (NanoDrop) and DNA yield was measured using the Qubit™ 3.0 Fluorometer (Thermo Fischer Scientific).

**Western Blot on B-lymphoblast samples**. Western Blot analysis was performed on HGPS patient B-lymphoblast cells obtained from the Coriell Institute (Affected HGPS patient; AG03506, unaffected sibling; AG03507). Fresh B-lymphoblast cell culture pellets were supplemented with 8 M UREA, 5% RIPA buffer including protease and phosphatase inhibitors (Roche, according to manufacturers guidelines) and homogenized by pipetting and vortexing. The protein lysates were briefly sonicated (2 min. at 4 °C on high amplitude) and the nuclear fraction was isolated using the Minute Nuclear Envelope Extraction Kit (NE-013). The isolated nuclear fractions were loaded on a SDS-PAGE gel (4% Mini-PROTEAN TGX Stain-Free, 456-8036 Bio-Rad) and transferred on a nitrocellulose membrane (1704158, Bio-Rad). The membrane was incubated with primary antibodies against lamin A/C (1:200, sc-376248 E-1, Santa Cruz Biotechnology) and the nuclear protein p84 (1:500, GTX70220 GeneTex) in 5% TBS-T milk over night at 4 °C. Secondary antibody staining was performed on the following day with a HRP-conjugated goat anti-mouse IgG antibody (1:10000, Jackson-ImmunoResearch).

**Western Blot on mouse skin samples**. Mouse skin proteins were extracted as previously described[33] and separated on an 8% SDS-Polyacrylamide gel. Protein separation by SDS-PAGE was performed for 1.5 h for β-actin and for 3 h for lamin A/C to achieve a better separation. Protein transfer was performed according to the standard procedure for the Semidry TransferCell (Bio-Rad). Primary antibodies: mouse monoclonal anti-human lamin A/C (1:200, mab3211, Chemicon) and β-actin (1:1000, A2228, Sigma–Aldrich) were used in 5% TBS-T milk over night at 4 °C. Secondary antibody staining was performed with an HRP-conjugated goat anti-mouse IgG antibody (1:10000, 115-035-062, JacksonImmunoResearch).

**Droplet-digital PCR rare event detection (ddPCR-RED)**. Rare event detection ddPCR was performed with the QX200 droplet digital PCR system (Bio-Rad). DdPCR assays for RED were designed for the *LMNA* c.1824C > T allelic variant and PCR thermal cycling conditions were optimized. The PCR reaction mix was prepared following the manufacturers guidelines (Bio-Rad) using a 2× ddPCR reaction Supermix for Probes (no dUTP, Bio-Rad), 20× ddPCR primer/probe mixture (FAM or HEX labelled, Bio-Rad), 5U of *HindIII* restriction enzyme (New England BioLabs) and template DNA. The polymerase in the 2× ddPCR reaction Supermix for probes (no dUTP, Bio-Rad) is able to read an inosine (I) nucleotide as a guanine (G) nucleotide and can therefore incorporate a cytosine (C) nucleotide in the newly synthesized strand (in agreement with Bio-Rad technical support). Raw data for each sample was analysed using the manufacturers software (QuantaSoft, version 1.6, Bio-Rad). Positive droplet amplification was classified if 3 or more droplets were detected in the measured FAM or HEX fluorescence channel. Fluorescence signal thresholds were defined according to a heterozygous DNA control sample. Each sample had to reach at least 10,000 droplets, with a minimum of 3 positive droplets in the target channel in order to be a valid quantification. The fractional abundance (FA) was then calculated from merging two valid technical replicates together.

The FA is calculated:

$$FA = \frac{\text{Number of alleles (A)}}{\text{Number of alleles (A)} + \text{Number of alleles (B)}} \qquad (1)$$

**Absolute quantification ddPCR (ddPCR-ABS)**. Absolute quantification ddPCR was performed using the QX200 ddPCR system (Bio-Rad). 800–1000 ng of RNA was extracted from B-lymphoblast cells or mouse tissue and cDNA was prepared. ABS-ddPCR was performed using the following primer sequences:

Krt15 Fwd: 5′-GAAGAGATCCGGGACAAAATTC-3′;
Krt15 Rev: 5′-CAGGGTCAGCTCATTCTCAT-3′;
RT LA only-Fwd: 5′-TCTTCTGCCTCCAGTGTCACG-3′;
RT LA only-Rev: 5′-AGTTCTGGGGGGCTCTGGGT-3′;
Gapdh Fwd: 5′-GAGCGAGATCCCTCCAAAAT-3′;
Gapdh Rev: 5′-CATCACGCCACAGTTTCC-3′;
ACTB Fwd: 5′-CCTAGGCACCAGGGTGTGAT-3′;
ACTBRev: 5′-CATGTCGTCCCAGTTGGTAA-3′;
Progerin Fwd: 5′-ACTGCAGCAGCTCGGGG-3′;
Progerin Rev: 5′-TCTGGGGGTCCTGGGC-3′.

Following PCR thermo cycling conditions were used: 5 min 95 °C, 40 cycles of 30 s 95 °C and 1 min 60 °C/60 °C/60 °C/63 °C/63 °C (ACTB, Gapdh, Krt15, and progerin, LMNA respectively). All samples were run in duplicates and analysed as previously described[54,55].

**Skin immunofluorescence**. Skin samples from LF-ABE and saline-treated mice were collected and fixed in 4% paraformaldehyde at 4 °C overnight, embedded in paraffin and cut into 4 µm sections. Sections were rehydrated, followed by heat-mediated antigen retrieval. Blocking was performed with 20% normalized goat serum combined with 1% BSA. After incubation overnight at 4 °C with primary antibodies, anti-progerin (1:100, ALX-804-662-R200, Enzo Life Science) and anti-53BP1 (1:1000, ab36823, Abcam) or anti-keratin 5 (1:500, 905501, BioLegend), anti-keratin 15 (1:1000, 833904, BioLegend) sections were incubated for 45 min at room temperature with the corresponding secondary antibody, AlexaFluor 488-conjugated goat anti-rabbit (1:500) and goat anti-chicken (1:500), AlexaFluor 555-conjugated goat anti-mouse (1:150). Sections were counterstained with DRAQ5 (1:1000, ThermoFischer Scientific) for 5 min prior to mounting (ProLong Gold antifade reagent, Molecular Probes). Confocal imaging was performed using a Nikon A1R and an A1+ imaging system (Nikon Corporation, Japan), with a 60× oil objective. Progerin protein and transcript analysis was performed by imaging the whole cross section of the injection site with one section per sample. In LF-ABE skin sections we then quantified an average number of 824 cells per section and for saline treated HGPS sections an average number of 881 cells per section in the interfollicular epidermis (IFE) for progerin protein quantification. For progerin transcript quantification we measured the number of transcripts appearing as dots per $mm^2$ across the whole IFE and in progerin negative cell clusters. Quantification was performed using the NIS elements analysis software (Nikon, v.5.3.04).

**Histological staining**. 4 µm-thick sections of overnight 4% FFPE saline and ABE treated HGPS skin were collected. Paraffin was removed by incubation of the section on 60 °C for one hour our and stained with Mayers Haematoxylin

(Histolab). Epidermal thickness measurements were performed using the NIS elements analysis software (Nikon, v.5.3.04) and the ImageJ software (1.53a). All haematoxylin pictures were obtained using a Nikon eclipse E-1000 microscope with a 10x magnification. Based on the scale bar we obtained the pixel to μm conversion (1 μm = 1,83 pixel, 10x magnification) in ImageJ and measured the epidermal thickness in a randomly selected region across the epidermis (35 consecutive measurements, same distance between each single measurement) for the LF-ABE treated samples and similar in saline treated samples (31 consecutive measurements, same distance between each single measurement).

**In situ hybridization**. Progerin and *Krt15* transcripts were detected using Base-Scope and RNAscope technologies respectively (Advanced Cell Diagnostic, Newark, CA), in accordance with the manufacturer's protocol. Few modifications were made: target retrieval was performed for 30 min and Protease III was applied for 15 min. After detection, blocking with 10% normal goat serum (NGS) was applied onto the slides, and sections were incubated with anti-progerin antibody (1:150, ALX-804-662-R200, Enzo Life Science) at 4 °C overnight in the dark. Thereafter, alexaFluor 647-conjugated goat anti-mouse antibody (1:500) was then added and nuclei were counterstained using DAPI (1:800). Confocal imaging was performed using a Nikon A1R and an A1+ imaging system (Nikon Corporation, Japan), with a 60× oil objective. Analysis was done using NIS elements software.

**Statistical analysis**. The statistical analyses were performed using unpaired *t* test with a two tailed 95% confidence interval and non-parametric one-way ANOVA analysis comparing multiple means with a 95% confidence interval. Graphs were plotted with GraphPad Prism (version 6.0). Data were presented as mean $+/-$ SEM (standard error of mean) in the text. $P < 0.05$ were considered significant.

**Reporting summary**. Further information on research design is available in the Nature Research Reporting Summary linked to this article.

## Data availability

The data supporting the findings from this study are available within the manuscript and its Supplementary Information. The DNA sequencing data has been deposited in the National Center For Biotechnology Information (NCBI) Sequence Read Archive (SRA) database with the BioProject accession code PRJNA744017. Source data are provided with this paper.

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

## Acknowledgements

This study was performed in part at the Live Cell Imaging Unit/Nikon Center of Excellence, Department of Biosciences and Nutrition, Karolinska Institutet, Huddinge, Sweden, and the Bioinformatics and Expression Analysis core facility at the Karolinska Institutet. The research was supported by grants to M.E. from the Torsten Söderberg stiftelse, DW from Karolinska Institutet faculty funding, the Swedish Research Council, the Center for Innovative Medicine, to G.R. from the Foundation for Geriatric Diseases at the Karolinska Institutet, Loo and Hans Osterman's Foundation for Medical Research, and to J.-S.K. from the Institute for Basic Science (IBS-R021-D1). Figure 2b, g, Supplementary Fig. 4b, adapted from "Large-sized syringe (1/2 liquid), Mouse (lateral 2)", by BioRender.com (2021). Retrieved from https://app.biorender.com/biorender-templates.

## Author contributions

D.W. had the original idea to use an ABE for the treatment of HGPS. D.W. and M.E. conceived and designed the study. K.L. and J.-S.K. designed and constructed the base editing tools. D.W., G.R., K.L., H.Y. and P.M. performed the experimental work. D.W., G.R., K.L., H.Y., M.B., P.M., J.-S.K. and M.E. analyzed and interpreted the data. D.W., K.Y. and G.R. performed the statistical analysis. D.W. wrote the first draft of the manuscript, and all authors critically revised the manuscript. J.-S.K. and M.E. obtained the funding and supervised the work. All authors read and approved the final manuscript.

## Funding

## Competing interests

J.-S.K. is a founder of and shareholder in ToolGen, Inc. The remaining authors declare no competing interests.
