## [Peer Review File · Nature Communications]

REVIEWER COMMENTS

Reviewer #1 (Remarks to the Author):

In a recent landmark study (ref. 26), the delivery of an adenine base editor (ABE) corrected the pathogenic HGPS mutation in human HGPS fibroblasts and in a LMNA c.1824 C>T transgenic mouse model. A single injection of AAV9 (IV) encoding the ABE resulted in widespread and extensive restoration of normal RNA splicing and reduced progerin protein levels in different tissues, including the skin. Importantly, lifespan increased more than twofold. However, concerns about safety have been raised for AAV vectors. Whisenant and coworkers reasoned that gene editing by a transient non-integrative delivery system might be a viable alternative. They tested this hypothesis in a skin-specific humanized HGPS mouse model where a human LMNA c.1824 C>T transgene is expressed in keratin 5 positive cells. The model develops skin disease phenotypes similar to children with HGPS. Injection of the ABE and sgRNA (IP) using an AAV5 vector system resulted in <0.5% correction efficiency in multiple tissues, including the skin. To examine the effects of adenine base editing in the skin, the authors performed ABE studies using a LentiFlash commercial vector system. The lentiviral particles were injected intradermally (in a volume of 50 μ l) into the dorsal skin at P21, and skin pathology was evaluated after four weeks. They report that the HGPS skin phenotype was improved and clusters of progerin-negative keratinocytes were detected, and conclude that transient non-integrative viral vector mediated ABE expression is a plausible approach for future gene editing therapies.

1. The authors injected saline as a control for the MS2-lentiviral delivery of ABE and sgRNA. A more appropriate experimental control is the injection of MS2 lentivirus with a non-targeting (NT) sgRNA.

2. The authors report that in saline or untreated HGPS mice, they develop severe skin phenotype with regional variations (lines 186-187). This suggests that the severity of the skin phenotype is variable at different regions. To minimize differences due to regional variation (and differences between mice), skin sites injected with ABE should be compared with adjacent control skin sites collected from the same mouse.

3. In Figure 3g, progerin and lamin A protein in skin samples were quantified with a human specific lamin A antibody and expressed relative to actin. However, in these mice, only keratin 5 positive cells express the human lamin A transgene, whereas keratin 5 positive and keratin 5 negative cells express actin. Thus, actin is not an appropriate loading control for human lamin A expressing cells. Since the human lamin A transgene also expresses GFP, a better approach would be to normalize

lamin A and progerin expression relative to GFP. Alternatively, progerin levels could be normalized to lamin A.

4. The authors report that progerin-negative cell clusters in LF-ABE treated mice contain an average of 8.6 cells per cluster. In saline treated controls, the average was 4.1 progerin-negative cells per cluster. What is the explanation for progerin-negative cell clusters in saline treated HGPS mice? Does this represent variation in tissue sampling or non-uniform expression of the LMNA transgene in progenitor cells?

Other comments:

1. Figure 1f. ABE treatment significantly reduced progerin transcript levels in HGPS lymphoblasts. If this is due to correction of the HGPS mutation, there should be an increase in lamin A transcripts. However, lamin A transcript levels are less than in untreated cells.

2. Figure 2d. Cells in the basal layer appear to express progerin transcripts, however, they do not express progerin protein. Is the expression of progerin protein variable in the skin or is the progerin probe used for the in situ hybridization studies not specific? Control studies to verify the specificity of the progerin probe should be presented (reported in the supplement).

3. The Methods require more experimental detail. Was the divided dose of virus injected at the same site, adjacent, or other? What was the size of the skin samples (diameter and depth) collected for the protein and RNA studies? How were progerin transcripts and protein expression quantified in skin samples (e.g., the number of sections examined/injection site).

4. For the general reader, it would be helpful to show a diagram of the inducible human HGPS/K5 bi-transgenic system. Also, it would be helpful to label the locations of the IFE and suprabasal layers in Figure 4.

5. In Figure 5c, the authors conclude that ABE treatment increases K15 expression in the basal layer, improving tissue homeostasis of the skin. However, quantification of K15 transcript levels was reported not to be significantly increased compared to saline (figures 5d, 5e).

Reviewer #2 (Remarks to the Author):

The current manuscript by Whisenant et al. examined the effect of adenine base editor (ABE) system in treating progeria skin caused by LMNA c.1824C>T mutation in a mouse model. They found that ABEmax-VQR could correct the mutation with low bystander editing by transfecting a B-lymphoblast cell line with LMNA c.1824C>T mutation. And the ABEmax-VQR system delivered by non-integrative lentiviral vector can correct 4.1% alleles in skin cells along with improved skin phenotype four weeks after intradermal injection. Recently Koblan et al. (Nature. 2021 Jan;589(7843):608-614) reported that a single injection of AAV9 encoding the ABEmax-VRQR resulted in substantial correction of the pathogenic LMNA c.1824C>T mutation (around 20–60% across various organs six months after injection), restoration of normal RNA splicing, reduction of progerin protein levels and double lifespan. In addition, they also reported corrected allele in skin was around 5% figured out from their Extended Data Fig.3d. Therefore, the novelty of the current manuscript is severely impaired. Cardiovascular complications and stroke are the most common causes of death of Hutchinson-Gilford progeria patients, so that skin is not a value target for treatment. What was the significance of targeting the skin in the progeria model?

Specific comments:

1. Line 142: The dosage “1011 GC/mL per mouse of AAV5” is unclear. The dosage unit should be GC per mouse or GC/g body weight. What was supported by reference 38 here?
2. Line 151: Fig.1a should be Fig.2a.
3. Line 171: Please use the standard “ \times ” instead of “x” in “4x10¹⁰” and the whole manuscript.
4. Line 193: Method for measuring epidermal thickness should be provided in method section.
5. Line 611 and 613: Please use the standard “ μ ” instead of “u”.
6. Line 621: How were the PCR products for sequencing generated? What was the sequencing depth?
7. Line 661: Authors stated that “For LF-ABE injections, offspring animals were then injected at P2 (IP) with 10uL of virus/g of body weight or at P21-22 (ID) with a total virus concentration of 1x10¹⁰ or 4x10¹⁰ total viral particles per mouse divided in two consecutive injections with 50uL each”. However, no P2 data was shown in the manuscript. And how was the ID injection performed specifically? Where was the injection site? How to collect the skin sample four weeks after injection?
8. Line 667: How to construct the split ABE system and what is the split site?
9. Line 672: How to titrate the viral particle?
10. Line 769 and 623: Please provide the full names of the different “NGS”.
11. Fig.1g: The sample for Western Blot was isolated nuclear fraction of B-lymphoblast cells, so that the reference protein should be a nuclear reference rather than β -actin.

12. Fig.5c: It is better to use immuofluorecence rather than in situ hybridization to demonstrate the expression of K15.

Reviewer #1 (Remarks to the Author):

In a recent landmark study (ref. 26), the delivery of an adenine base editor (ABE) corrected the pathogenic HGPS mutation in human HGPS fibroblasts and in a LMNA c.1824 C>T transgenic mouse model. A single injection of AAV9 (IV) encoding the ABE resulted in widespread and extensive restoration of normal RNA splicing and reduced progerin protein levels in different tissues, including the skin. Importantly, lifespan increased more than twofold. However, concerns about safety have been raised for AAV vectors. Whisenant and coworkers reasoned that gene editing by a transient non-integrative delivery system might be a viable alternative. They tested this hypothesis in a skin-specific humanized HGPS mouse model where a human LMNA c.1824 C>T transgene is expressed in keratin 5 positive cells. The model develops skin disease phenotypes similar to children with HGPS. Injection of the ABE and sgRNA (IP) using an AAV5 vector system resulted in <0.5% correction efficiency in multiple tissues, including the skin. To examine the effects of adenine base editing in the skin, the authors performed ABE studies using a LentiFlash commercial vector system. The lentiviral particles were injected intradermally (in a volume of 50 µl) into the dorsal skin at P21, and skin pathology was evaluated after four weeks. They report that the HGPS skin phenotype was improved and clusters of progerin-negative keratinocytes were detected, and conclude that transient non-integrative viral vector mediated ABE expression is a plausible approach for future gene editing therapies.

1. The authors injected saline as a control for the MS2-lentiviral delivery of ABE and sgRNA. A more appropriate experimental control is the injection of MS2 lentivirus with a non-targeting (NT) sgRNA.

Author reply: We agree with the reviewer that it is important to use appropriate controls. However, we do not think that addition of the NT-sgRNA control group would be essential in this instance. NT-sgRNA are mainly used as a negative control as a reference for sgRNA library screening experiments, for example to detect specific loss-of-function mutations (Sanjana, et al, Nature Methods, 11, 783-84 (2014)), or to confirm the sgRNA-dependent activities of newly developed genome engineering tools (Özcan, Ahsen, et al, Nature, 1-6

(2021)). Previously published papers using in vivo genome editing have used saline-injected or untreated mice as controls, and that is the reason why we chose to use the same experimental control. (Koblan, et al, Nature 589, 608-614 (2021); Wilkinson, et al., Nature Communications 12,686 (2021); Ryu, et al. Nature Biotechnology, 36, 536-39 (2018); Kim, et al, Nature Communications 8, 14500 (2017), Musunuru et al., Nature 593, 429-434 (2021), Rothgangl et al., Nature Biotechnology 39, 949-957 (2021)). In addition, Koblan et al. have shown the efficiency of the c.1824C>T targeting sgRNA and our data confirms that the same c.1824C>T targeting sgRNA has a successful base editing activity (Fig.1b, d-f; Fig.2b-e).

2. The authors report that in saline or untreated HGPS mice, they develop severe skin phenotype with regional variations (lines 186-187). This suggests that the severity of the skin phenotype is variable at different regions. To minimize differences due to regional variation (and differences between mice), skin sites injected with ABE should be compared with adjacent control skin sites collected from the same mouse.

Author reply: Thank you for this comment. In the revised version we have included data from adjacent skin control samples (Fig.3c, e-f). These skin control samples were taken approximately 2 cm away from the intradermal LF-ABE injection site and scored by the same pathologist. We did not observe significant differences between the saline control samples and the adjacent skin control samples. The text has been revised accordingly.

3. In Figure 3g, progerin and lamin A protein in skin samples were quantified with a human specific lamin A antibody and expressed relative to actin. However, in these mice, only keratin 5 positive cells express the human lamin A transgene, whereas keratin 5 positive and keratin 5 negative cells express actin. Thus, actin is not an appropriate loading control for human lamin A expressing cells. Since the human lamin A transgene also expresses GFP, a better approach would be to normalize lamin A and progerin expression relative to GFP. Alternatively, progerin levels could be normalized to lamin A.

Author reply: We agree with the reviewer that GFP would be a more appropriate control. We have tested three antibodies for GFP on western blot (TaKaRa 632460, Abcam 6673

and Abcam 290) but we could not obtain reliable results. Therefore we have normalized progerin levels to lamin A, in agreement with the suggestion from the reviewer (see figure 3i).

4. The authors report that progerin-negative cell clusters in LF-ABE treated mice contain an average of 8.6 cells per cluster. In saline treated controls, the average was 4.1 progerin-negative cells per cluster. What is the explanation for progerin-negative cell clusters in saline treated HGPS mice? Does this represent variation in tissue sampling or non-uniform expression of the LMNA transgene in progenitor cells?

Author reply: We have not found evidence for variation in tissue sampling or non-uniform expression of the *LMNA* transgene in progenitor cells in this model. The epidermis is known to also contain other cell types besides keratinocytes. These cell types include langerhans cells, merkel cells, dendritic epidermal T cells and melanocytes (Reviewed by Fuchs and Blau et al., *Cell Stem Cell*, 27, 532-556 (2020)). The Keratin 5 promotor that was used to drive the expression of the lamin A minigene transgene in this model is active in epidermal keratinocytes (Diamond et al. *J Invest Dermatol*, 115, 788-794 (2000)).

Other comments:

1. Figure 1f. ABE treatment significantly reduced progerin transcript levels in HGPS lymphoblasts. If this is due to correction of the HGPS mutation, there should be an increase in lamin A transcripts. However, lamin A transcript levels are less than in untreated cells.

Author reply: Just to clarify; Figure 1f shows that ABE treatment affects progerin transcript levels in HGPS lymphoblasts, but this is not supported by statistical testing. Neither are the lamin A transcript levels significantly reduced (Fig. 1f). In figure 3g (previous version figure 3f) we show that progerin transcripts are significantly lower upon in vivo ABE treatment, and that the lamin A transcript levels are unchanged. This is in agreement with a previous study that used absolute transcript quantification of lamin A in HGPS patient cells and showed that lamin A transcripts were not lower in HGPS patients compared to unaffected controls. This study and our in vivo data suggest that the levels of lamin A

transcripts are unaffected in the presence of the c.1824C>T mutation, and independent of progerin transcript levels (Rodriguez et al., *Eur. J. Hum. Genet.*,17:928-37 (2009)). We have now addressed this in the result section..

2. Figure 2d. Cells in the basal layer appear to express progerin transcripts, however, they do not express progerin protein. Is the expression of progerin protein variable in the skin or is the progerin probe used for the in situ hybridization studies not specific? Control studies to verify the specificity of the progerin probe should be presented (reported in the supplement).

Author reply: Following the reviewer's recommendation, we now provide data on the specificity of our progerin probe for in situ hybridization in the revised manuscript. Using HGPS mouse skin (K5tTA/tetop-LA^{G608G}) as a positive control and mouse skin expressing human lamin A only (K5tTA/tetop-LA^{WT}) (Sagelius et al., 121, 969-78 (2008)) as a negative control, we show that the progerin probe specifically detects progerin and does not recognize mouse/human lamin A or any other transcripts (Supplementary fig.6a,b). Our results suggest that the expression levels of the progerin protein is variable, possibly caused by its accumulation.

3. The Methods require more experimental detail. Was the divided dose of virus injected at the same site, adjacent, or other? What was the size of the skin samples (diameter and depth) collected for the protein and RNA studies? How were progerin transcripts and protein expression quantified in skin samples (e.g., the number of sections examined/injection site).

Author reply: We thank the reviewer for noticing this omission. We have included more experimental detail in the methods part of the revised manuscript and tried to answer the questions of the reviewer.

Was the divided dose of virus injected at the same site, adjacent, or other?

Author reply: The divided dose (1×10^{10} vps or 4×10^{10} vps) was injected with a volume of 2x 50 μ l at the same site. We used a marker to indicate the injection site.

What was the size of the skin samples (diameter and depth) collected for the protein and RNA studies?

Author reply: For the protein and RNA studies the size of the skin samples was of 3-5 mm in diameter and 1-2 mm in depth. Skin samples were collected at the marked injection site.

How were progerin transcripts and protein expression quantified in skin samples (e.g., the number of sections examined/injection site)

Author reply: For progerin transcript and protein expression quantification in skin samples, we examined the cross-section of the whole injection site. Here we examined 1 section per mouse and scanned the whole section to visualize progerin negative cell clusters. For progerin protein expression, we quantified an average number of 881 IFE cells for each of the saline-treated samples (n=5) and an average number of 824 IFE cells for each of the LF-ABE treated samples (n=5). For progerin transcript quantification, we measured the number of transcripts appearing as dots per mm² across the whole IFE and in progerin negative cell clusters.

4. For the general reader, it would be helpful to show a diagram of the inducible human HGPS/K5 bi-transgenic system. Also, it would be helpful to label the locations of the IFE and suprabasal layers in Figure 4.

Author reply: We have now included a schematic drawing of the inducible human HGPS/K5 bi-transgenic system (Fig.2a) and included the labels for the IFE basal and suprabasal layers in the immunofluorescence pictures in Figure 4.

5. In Figure 5c, the authors conclude that ABE treatment increases K15 expression in the

basal layer, improving tissue homeostasis of the skin. However, quantification of K15 transcript levels was reported not to be significantly increased compared to saline (figures 5d, 5e).

Author reply: We apologize for this mistake. We did observe a significant increase of K15 transcripts ($P=0,04$) but the asterisk (*) indicating significance index in figure 5d was not included. In situ hybridization allowed the detection of K15 transcripts specifically in basal cells (Fig 5d), while qRT-ddPCR allowed for the detection of K15 transcripts in the whole skin tissue (since RNA was extracted from the skin) (Fig 5e), which could explain the discrepancy seen between Fig 5d and 5e. Since the more refined analysis of K15, that focused on the basal keratinocyte layer, showed a significant difference we decided to refer to this data and state that we observed a significant difference. In the revised version of the manuscript we have now tried to clarify this.

Reviewer #2 (Remarks to the Author):

The current manuscript by Whisenant et al. examined the effect of adenine base editor (ABE) system in treating progeria skin caused by LMNA c.1824C>T mutation in a mouse model. They found that ABEmax-VQR could correct the mutation with low bystander editing by transfecting a B-lymphblast cell line with LMNA c.1824C>T mutation. And the ABEmax-VQR system delivered by non-integrative lentiviral vector can correct 4.1% alleles in skin cells along with improved skin phenotype four weeks after intradermal injection. Recently Koblan et al. (Nature. 2021 Jan;589(7843):608-614) reported that a single injection of AAV9 encoding the ABEmax-VRQR resulted in substantial correction of the pathogenic LMNA c.1824C>T mutation (around 20–60% across various organs six months after injection), restoration of normal RNA splicing, reduction of progerin protein levels and double lifespan. In addition, they also reported corrected allele in skin was around 5% figured out from their Extended Data Fig.3d. Therefore, the novelty of the current manuscript is severely impaired. Cardiovascular complications and stroke are the most common causes of death of Hutchinson-Gilford progeria patients, so that skin is not a

value target for treatment. What was the significance of targeting the skin in the progeria model?

Authors reply: We agree with the reviewer about the importance of the study by Liu and colleagues. However, we believe that our study provides additional insights.

First, the study by Liu et al. used a two-copy transgenic mouse model. This means that 2 copies of the mutant allele were present in every cell. When the base editor enters the cell it is expected to correct both alleles. This means that the number of cells that are corrected is actually half of the allele frequency (10-30%, and in the skin 2.5% of the cells). In this study we have used a one-copy transgenic mouse model, and the fractional abundance that we obtained using a transient system led to mutation correction in 24.1% and 20.8% of the skin cells (range 13-43.8%). Four weeks after the initial delivery of the ABE we had an editing frequency of 4.1% indicating progenitor cell editing and we observed an improved skin phenotype. This suggests that repeated exposures of non-genotoxic, transient base editors could be used to treat genetic syndromes.

Second, the study by Koblan et al. used an AAV system to deliver the base editors. Recent studies report safety concerns with the use of AAVs. In this study we have used a transient non-integrative delivery system to deliver the base editors and show promising results.

Third, when analyzing the bystander editing of the A10 position obtained from the sgRNA, we identified a potential new target for future gene editing approaches since editing of this specific nucleotide is predicted to reduce progerin splicing below the reference sequence.

We have revised the text and tried to better emphasize the additional insights obtained from our study compared to the Koblan et al.

We chose to target the skin as it is a central HGPS phenotype and is affected in all HGPS patients.

Specific comments:

1. Line 142: The dosage "1011 GC/mL per mouse of AAV5" is unclear. The dosage unit should be GC per mouse or GC/g body weight. What was supported by reference 38 here?

Author reply: We have changed the dosage description to 10^{11} GC per mouse (Line 142) as suggested by the reviewer. The reference 38 (Line 142) is supporting the tropism of the AAV5 vector for dermal fibroblasts and keratinocytes. This was the reason for the selection of AAV5 in our first experiments using base editors in HGPS mice.

2. Line 151: Fig.1a should be Fig.2a.

Author reply: We thank the reviewer for noticing this mistake, which we corrected in the revised version of the manuscript.

3. Line 171: Please use the standard "x" instead of "x" in "4x10¹⁰" and the whole manuscript.

Author reply: In the revised manuscript we have corrected the standard "X" to "x" in 4×10^{10} and in the entire manuscript.

4. Line 193: Method for measuring epidermal thickness should be provided in method section.

Author reply: As suggested by the reviewer, we have in the revised manuscript included the description of how we measured the epidermal thickness under the method section: Histological staining.

5. Line 611 and 613: Please use the standard "μ" instead of "u".

Author reply: We have corrected and revised the line 611 and 613 and replaced the u with the standard "μ" in the revised version of the manuscript.

6. Line 621: How were the PCR products for sequencing generated? What was the sequencing depth?

Author reply: We thank the reviewer for noticing this omission. In the revised version we have included a new section in the Methods that reads as follow:

"Targeted deep sequencing of genomic DNA:

Genomic DNA was isolated from sorted cells using a DNeasy Blood & Tissue Kit (Qiagen) at 72 hours after transfection according to manufacturer's protocol. Preparation of sequencing library and NGS running were performed as described previously³⁶. In brief, the region of interest was first amplified from genomic DNA using KAPA HiFi HotStart DNA polymerase (Roche) to a size of ~500bp. Then, amplicons were subjected to PCR amplification to label each fragment with index and adapter sequences for TruSeq DNA-RNA CD index system (Illumina). The sequencing libraries were purified using a PCR purification kit (MGmed) and sequenced using a MiniSeq (Illumina) with an average read depth of ~5000. Primer sequences used for targeted deep sequencing at on- and off-target sites are listed in Supplementary Table 2. Substitution frequencies at each base position and allelic fractions were measured using BE-analyzer⁵⁰. Indel frequencies were measured using Cas-analyzer⁵¹."

7. Line 661: Authors stated that "For LF-ABE injections, offspring animals were then injected at P2 (IP) with 10uL of virus/g of body weight or at P21-22 (ID) with a total virus concentration of 1×10^{10} or 4×10^{10} total viral particles per mouse divided in two consecutive injections with 50uL each". However, no P2 data was shown in the manuscript. And how was the ID injection performed specifically? Where was the injection site? How to collect the skin sample four weeks after injection?

Author reply: In the revised version we have corrected the sentence to: "For LF-ABE injections, offspring animals were injected at P21-22 (ID) with a total virus concentration of 1×10^{10} or 4×10^{10} total viral particles per mouse divided in two consecutive injections with 50uL each."

In reply to how the ID injection was performed specifically, we have included a section about the injection procedure and tissue sampling in the methods part:

"Intradermal injection and skin collection:

All animals were injected using 0,3mL × 8mm (30G) U-100 insulin syringes (BD, MicroFine) under isoflurane anesthesia. On the first day of the injection period, LF-ABE viral particle aliquot solutions of 50µL were thawed on ice and loaded into the syringes and injected. On the second day (P22) this process was repeated and the viral particle solution was injected

at the exact same site of the previous injection. The injection site was located on the right lower back in all saline and LF-ABE injected animals. All hair was removed by shaving before the injection and the skin area was cleaned with 70% ethanol. After the intradermal injection was performed, the injected skin region was circled with a surgical marker. This procedure was repeated on the following day. All animals were monitored daily and the marking of the injection site was updated twice per week with a removal of regrown hair by shaving. Either 2 days after the first injection or at 4 weeks after the initial injection (p21), animals were sacrificed. In order to exclude dilution and/or variation effects we took care to collect the same size of the skin tissue sample from different animals. The skin of the marked injection site was extracted in a circular shape (diameter \approx 1cm, depth \approx 1-2mm) using scissors and forceps and divided. One half of the circular sample was used for DNA, RNA and protein extraction (diameter \approx 3-5mm, depth \approx 1-2mm), the other half was preserved in 4% paraformaldehyde (PFA) for staining.”

8. Line 667: How to construct the split ABE system and what is the split site?

Author reply: We apologize for overlooking this. In the revised version we have included a new section in the Methods:

“Custom adeno-associated virus serotype 5 (AAV5) production:

Trans-splicing AAV vectors encoding sgRNA and split ABE7.10-VQR (at Gly315-Gln316 of Cas9) were generated using NEBuilder HiFi DNA Assembly Master Mix (New England Biolabs) to insert EF1 α promoter and sgRNA sequences into the pAAV-ABE-NT-sgRNA plasmid (addgene #112734, Ref. 27 [Ryu, S.M. et al]) and to incorporate mutations in the Cas9 PAM-interacting domain (D1135V/R1335Q/T1337R) into the pAAV-ABE-CT plasmid (addgene #112876, Ref. 27 [Ryu, S.M. et al]). Trans-splicing AAV vectors were sent to VigeneBioscience and packaged into AAV5 viral particles (500mL, 1×10^{13} GC, for each packaging). Titrated and ready to use ABE-NT and ABE-CT AAV5 vectors were obtained and injected into HGPS mice.”

9. Line 672: How to titrate the viral particle?

Author reply: In the revised version we have included a description of how the titration was done in the Method section:

“Titers are obtained by a P24 ELISA (Vectalys). The p24 core antigen is detected directly on the viral supernatant with a HIV-1 p24 ELISA kit provided by Perkin Elmer. The captured antigen is complexed with a biotinylated polyclonal antibody to HIV-1 p24, followed by a streptavidin-HRP (horseradish peroxidase) conjugate. The resulting complex is detected by incubation with ortho-phenylenediamine-HCL (OPD), which produces a yellow color that is directly proportional to the amount of p24 captured. The absorbance of each microplate well is determined using a microplate reader and calibrated against the absorbance of an HIV-1 p24 antigen standard curve.”

10. Line 769 and 623: Please provide the full names of the different “NGS”.

Author reply: Following the reviewer’s suggestion, the full name of the different “NGS” was included and it now reads “targeted deep sequencing” and “normal goat serum” respectively.

11. Fig.1g: The sample for Western Blot was isolated nuclear fraction of B-lymphoblast cells, so that the reference protein should be a nuclear reference rather than β -actin.

Author reply: We appreciate this comment from the reviewer. As suggested, we have now included P84 as a nuclear matrix reference protein instead of β -actin, Figure 1g.

12. Fig.5c: It is better to use immunofluorescence rather than in situ hybridization to demonstrate the expression of K15.

Author reply: As suggested by the reviewer, we have included results from immunofluorescence from skin of saline and ABE-treated mice using an antibody to keratin 15 protein (Fig. 5f-g).

REVIEWER COMMENTS

Reviewer #1 (Remarks to the Author):

Thank you for your replies to my questions.

Question 4. The authors' explanation suggests that the detection of progerin-negative cells in saline treated HGPS mice is due to contamination of cell types that do not express keratin 5 (Langerhans cells, merkel cells, dendritic epidermal T cells and melanocytes). If this is correct, the authors should report the results as "negative cells per keratin 5-positive cells." The authors already have the data, and will not require additional experiments.

Other comments #1. The authors' explanation that absolute levels of progerin and lamin A transcripts were measured in figure 3g is reasonable. This should be added to the legend to figure 3g.

However, I am puzzled by the authors' response, "Just to clarify; Figure 1f shows that ABE treatment affects progerin transcript levels in HGPS lymphoblasts, but this is not supported by statistical testing." This appears inconsistent with the description in Results (p. 6), "Next, we analysed progerin and lamin A transcript levels and detected an average reduction of progerin transcripts of 83.8% (mean value of 1:1 and 3:1 ABE_{max}-VQR:sgRNA mass ratio) in edited HGPS patient cells (Fig.1f)." Please clarify.

Reviewer #2 (Remarks to the Author):

The authors have addressed most of my comments. There are some suggestions:

- 1.It is recommended that gene symbols should be corrected according to gene nomenclature rules established by HUGO.
- 2.Please provide quantification data of Figure 1g as shown in Figure 3i.
- 3.The author argued that the transcription level of Lamin A is not correlated with progerin mRNA level. mRNA of progerin reduced significantly in Figure 3g compared to Lamin A. However, no change

of Progerin/LaminA could be found in protein level, as shown in Figure 3i. It appeared that the rescue of phenotype was contributed by the reduction of both Lamin A and Progerin Figure 3a-f and 3h. What is the explanation for this phenomenon?

4. After two days of treatment, LF-ABE showed a high editing ratio in the skin. The author argued that this accounted for A-to-I editing in the early stages. One possible hypothesis could be that sequences containing inosine may not favour PCR processes and tend to be less amplified in conventional PCR reactions. Is there any difference in amplification behaviours between strands containing G and I in ddPCR?

REVIEWER COMMENTS

Reviewer #1 (Remarks to the Author):

Thank you for your replies to my questions.

Question 4. The authors' explanation suggests that the detection of progerin-negative cells in saline treated HGPS mice is due to contamination of cell types that do not express keratin 5 (Langerhans cells, merkel cells, dendritic epidermal T cells and melanocytes). If this is correct, the authors should report the results as "negative cells per keratin 5-positive cells." The authors already have the data, and will not require additional experiments.

Author reply: Thank you for this comment. Yes, this is what we suggested. We have now re-analyzed the data according to the reviewer's suggestion and report the results as "Progerin-negative per keratin 5-positive cells/cluster" (Fig.4c and Supplementary Fig.5) The manuscript text has been rephrased accordingly. The results show that in saline treated animals, on average 98,4% of the keratin 5-positive cells are also progerin-positive (using this specific progerin antibody).

Other comments #1. The authors' explanation that absolute levels of progerin and lamin A transcripts were measured in figure 3g is reasonable. This should be added to the legend to figure 3g.

Author reply: We have added the following sentence to the legend of Figure 3g as suggested by the reviewer: "Absolute quantification (ddPCR-ABS) was used to measure progerin and lamin A transcript levels in LF-ABE and saline-treated HGPS mouse skin four weeks post-treatment."

However, I am puzzled by the authors' response, "Just to clarify; Figure 1f shows that ABE treatment affects progerin transcript levels in HGPS lymphoblasts, but this is not supported by statistical testing." This appears inconsistent with the description in Results (p. 6), "Next, we analysed progerin and lamin A transcript levels and detected an average reduction of progerin transcripts of 83.8% (mean value of 1:1 and 3:1 ABEmax-VQR:sgRNA mass ratio) in edited HGPS patient cells (Fig.1f)." Please clarify.

Author reply: We apologize for this inconsistency. We have rephrased the sentence as follow: "Next, we analysed progerin and lamin A transcript levels and detected both transcripts in edited HGPS patient cells (Fig.1f)." In addition, we have revised Figure 1f so that there is no suggestion of significant reduction and to avoid confusion, since it is not recommended to perform statistical analysis on N = 2.

Reviewer #2 (Remarks to the Author):

The authors have addressed most of my comments. There are some suggestions:

1.It is recommended that gene symbols should be corrected according to gene nomenclature rules established by HUGO.

Author reply: Following the reviewer's recommendation we have revised the manuscript and tried to correct the gene nomenclature according to the HUGO nomenclature rules. Specifically we have revised the manuscript to use the appropriate HUGO nomenclature for *Krt5* and *Krt15* in the text and on the axis labeling of the graphs.

In regards to the *LMNA* gene, we think it would be misleading to "replace "lamin A and progerin transcripts" by *LMNA*". The *LMNA* gene encodes for both lamin A and C, but the transgene used here does not express lamin C. Therefore, we have decided to keep the old version with lamin A and progerin transcripts instead of referring to the *LMNA* gene.

2. Please provide quantification data of Figure 1g as shown in Figure 3i.

Author reply: We have performed data quantification of Figure 1g as requested by the reviewer. This data has been included in the revised version as Supplementary Figure 1d.

3. The author argued that the transcription level of Lamin A is not correlated with progerin mRNA level. mRNA of progerin reduced significantly in Figure 3g compared to Lamin A. However, no change of Progerin/LaminA could be found in protein level, as shown in Figure 3i. It appeared that the rescue of phenotype was contributed by the reduction of both Lamin A and Progerin Figure 3a-f and 3h. What is the explanation for this phenomenon?

Author reply: We have no clear explanation for this. We were surprised by the results of the progerin transcript quantification (almost 50% drop, as seen in Figure 3g) and that it did not reflect the frequency of T>C mutation correction detected in the skin samples at the same time point (4.1%, as measured by targeted deep-sequencing). A possible explanation for this observation could be that the *Krt5* promoter has different activity in different cells, and that the mutation correction occurred in cells that have higher expression of progerin. While the *Krt5* promoter is active in cells of the basal layer of the interfollicular epidermis, we also see expression of the *Krt5* gene in suprabasal cells of the interfollicular epidermis in this progeria skin model. The discrepancy with no detected reduction of progerin protein relative to lamin A, as measured by Western blot, might be masked by the remaining hyperplastic phenotype and accumulation of protein in the suprabasal layers.

4. After two days of treatment, LF-ABE showed a high editing ratio in the skin. The author argued that this accounted for A-to-I editing in the early stages. One possible hypothesis could be that sequences containing inosine may not favour PCR processes and tend to be less amplified in conventional PCR reactions. Is there any difference in amplification behaviours between strands containing G and I in ddPCR?

Author reply: We agree with this hypothesis. It has previously been shown that conventional PCR can have a reduced polymerase amplification efficiency for Inosine. In the revised version, we have included references to this work (ref 39, 40). In addition, our data shows a loss of possible A-to-I edited (c.1824C) alleles when we performed regular PCR before the ddPCR analysis (Supplementary Fig.4a-c). The ddPCR manufacturer (Bio-Rad) confirmed that the polymerase included in the ddPCR assay amplifies templates containing Inosine bases. This is included in the materials and methods part of the manuscript. Bio-Rad does not provide which polymerase they include in the ddPCR assay reaction, as well as possible differences in amplification behaviors between templates containing G and I bases.

REVIEWERS' COMMENTS

Reviewer #2 (Remarks to the Author):

I have no further comments.

REVIEWERS' COMMENTS:

Reviewer #2 (Remarks to the Author):

I have no further comments.

Author reply:

Thank you for your comments.